# Deep learning-based image analysis identifies a DAT-negative subpopulation of dopaminergic neurons in the lateral Substantia nigra

Nicole Burkert[1,9], Shoumik Roy [1,9✉], Max Häusler[1,9], Dominik Wuttke[2], Sonja Müller[1], Johanna Wiemer[1], Helene Hollmann[1], Marvin Oldrati[1], Jorge Ramirez-Franco [3,4], Julia Benkert[1], Michael Fauler[5], Johanna Duda[1], Jean-Marc Goaillard[3,4], Christina Pötschke[1], Moritz Münchmeyer[2,6], Rosanna Parlato[1,8] & Birgit Liss [1,7✉]

Here we present a deep learning-based image analysis platform (DLAP), tailored to autonomously quantify cell numbers, and fluorescence signals within cellular compartments, derived from RNAscope or immunohistochemistry. We utilised DLAP to analyse subtypes of tyrosine hydroxylase (TH)-positive dopaminergic midbrain neurons in mouse and human brain-sections. These neurons modulate complex behaviour, and are differentially affected in Parkinson's and other diseases. DLAP allows the analysis of large cell numbers, and facilitates the identification of small cellular subpopulations. Using DLAP, we identified a small subpopulation of TH-positive neurons (~5%), mainly located in the very lateral Substantia nigra (SN), that was immunofluorescence-negative for the plasmalemmal dopamine transporter (DAT), with ~40% smaller cell bodies. These neurons were negative for aldehyde dehydrogenase 1A1, with a lower co-expression rate for dopamine-D2-autoreceptors, but a ~7-fold higher likelihood of calbindin-d28k co-expression (~70%). These results have important implications, as DAT is crucial for dopamine signalling, and is commonly used as a marker for dopaminergic SN neurons.

[1] Institute of Applied Physiology, Medical Faculty, Ulm University, 89081 Ulm, Germany. [2] Wolution GmbH & Co. KG, 82152 Munich, Germany. [3] UMR_S 1072, Aix Marseille Université, INSERM, Faculté de Médecine Secteur Nord, Marseille, France. [4] INT, Aix Marseille Université, CNRS, Campus Santé Timone, Marseille, France. [5] Institute of General Physiology, Medical Faculty, Ulm University, 89081 Ulm, Germany. [6] Department of Physics, University of Wisconsin-Madison, Madison, WI, USA. [7] Linacre College & New College, Oxford University, OX1 2JD Oxford, UK. [8] Present address: Division of Neurodegenerative Disorders, Department of Neurology, Medical Faculty Mannheim, Mannheim Center for Translational Neurosciences, Heidelberg University, 68167 Mannheim, Germany. [9] These authors contributed equally: Nicole Burkert, Shoumik Roy, Max Häusler. ✉email: shoumik.roy@uni-ulm.de; birgit.liss@uni-ulm.de

Dopamine releasing (DA) neurons within the midbrain are important for a variety of brain functions and behavioural processes like voluntary movement control, learning, cognition, and motivation[1–4]. The cell bodies of DA midbrain neurons are predominantly located in two overlapping nuclei, the Substantia nigra pars compacta (SN) and the ventral tegmental area (VTA), with axonal projections to the dorsal striatum, or the ventral striatum and prefrontal cortex, respectively[5–8]. In line with these differential functions and projections, dysfunction of the dopaminergic midbrain system can lead to severe diseases like Parkinson's, Schizophrenia, addiction, or attention deficit hyperactivity disorders, ADHD[9–12]. Importantly, not all DA midbrain neurons are affected equally in these diseases. More precisely, in Parkinson's disease (PD), the second most common neurodegenerative disorder, a subpopulation of mesostriatal SN DA neurons is progressively degenerating, while neighbouring VTA DA neurons remain largely intact[13–16]. In contrast, in Schizophrenia, mesocorticolimbic VTA DA neurons display complex dysfunctions, while SN DA neurons are mainly unaffected[17–19]. The cause for this differential vulnerability of dopaminergic midbrain neurons is still unclear, and only unspecific, symptomatic therapies are available, with their respective side-effects[20,21].

One prerequisite for curative and cell-type specific therapies is a better understanding of the distinct subpopulations of dopaminergic neurons and their selective pathophysiology[5,8]. Analysis of mRNA and protein expression with single cell resolution and its correlation with anatomical locations and projections in health and disease states is an essential approach for identifying molecular determinants of differential neuronal vulnerability[22–26]. For this, labelling of mRNA or proteins in tissue sections, followed by microscopic imaging, are important tools[27–29]. However, manual neuronal mapping, cell counting, and quantification of mRNA- and protein-derived fluorescence signals is very time consuming and prone to human error[30–32]. Here, we present a deep learning-based image analysis platform (DLAP) that overcomes these issues. Our single cell DLAP approach and six individual algorithms were tailored to quantify autonomously (i) the number of distinct cell types in defined areas, as well as (ii) fluorescence signals, derived from either RNAscope probes (mRNA) or immunohistochemistry (proteins), in defined compartments - more precisely in plasma-membranes, cell body/cytoplasm, and nuclei of individual neurons. We utilised artificial neuronal networks (ANN) based artificial intelligence/deep learning approaches for image analysis. ANN are biologically inspired computer programs, designed to simulate the way in which the human brain processes information. ANN gather their knowledge by detecting the patterns and relationships in data (segmentation), and learn (or are trained) through experience, not by programming[33–36]. ANN consist of artificial neurons, arranged in different layers[37,38], where each consecutive layer obtains inputs from its preceding layer. As only the first (input) and the last (output) layer are visible/accessible, all layers in between are referred to as hidden layers. Network architectures containing hundreds of hidden layers are called deep networks[33]. After suitable training, deep learning networks can effectively extract relevant cellular features for automated image analysis[39].

To facilitate easy adaption to the most common respective tasks in life science, we have predesigned six algorithms, based on two distinct network structures, DeepLab3 (DL3) and fully convolutional neural networks (FCN)[40,41]. We trained them to detect, count, and analyse individual DA midbrain neurons, labelled for tyrosine hydroxylase (TH), the key-enzyme for synthesis of dopamine and other catecholamines[42]. However, the flexible design of the algorithms and the workflow allows quick and easy adaptions to other cell types and specimens, and thus is of general interest. Here, we detail the DLAP pipeline, including all six predesigned algorithms for automated image processing. As proof-of-principle, we demonstrate its reliable identification and count of cells in defined areas, as well as fluorescence-signal quantification in cellular compartments, by analysing about 40.000 TH-positive midbrain neurons. Using this approach for quantification of the immunofluorescence signal of the plasma-lemmal dopamine transporter (DAT) in TH-positive SN neurons, we identified a small subpopulation of DAT immuno-negative neurons (~5%), mainly located in the caudo-lateral parts of the SN (~37% of all lateral TH-positive SN neurons). These neurons had ~40% smaller cell bodies and also showed a co-expression profile for three additional markers of subpopulations of SN DA neurons (dopamine-D2-autoreceptor, $Ca^{2+}$ binding protein calbindin-d28k, and aldehyde dehydrogenase 1) that is rather untypical for classical SN DA neurons. The DAT is an electrogenic symporter of the $Na^+/Cl^-$ dependent transporter family (SLC6). It is crucial for re-uptake of dopamine at somatodendrites and axons, and also mediates an electric conductance[43]. The identification of a DAT-negative TH-positive SN neuron population has not only physiological relevance[44–46], but it has additional implications, as DAT is commonly used as marker for SN DA neurons and for their specific targeting, e.g., to generate transgenic mice, expressing Cre-recombinase under the control of the DAT-promoter[47–49].

## Results

Based on the Wolution web-interfaces (https://wolution.ai/), we developed a deep learning-based, automated image analysis platform (DLAP) as well as six tailored algorithms (two based on FCN, four based on DL3) for fast and reliable identification and quantitative analysis of TH-positive midbrain neurons within the SN and VTA of mouse and human *post mortem* brain sections. These six DLAP algorithms were trained by hundreds of manually marked TH- and DAPI-positive cell bodies of midbrain neurons. Figure 1 illustrates the general workflow. Tables 1, 2 and S1 provide details of algorithms, trainings, and performances (further details in methods and supplement). For all approaches, the region of interest (ROI) for cell type identification and quantification (i.e the SN) was marked manually, according to anatomical landmarks[50]. All six algorithms allowed reliable identification of target cells, as well as their quantification in a given ROI (DLAP-3 & 4), and/or quantification of mRNA-derived (DLAP-1 &2) or protein-derived (DLAP-5 & 6) fluorescence signals of distinct genes of interest in these target cells - all common tasks in life science.

Deep learning networks for image segmentation of complicated scenes, such as the Pascal Visual Object Classes (VOC) dataset[51], that we used in the DeepLab3 based algorithms, are usually very large, and their filters are particularly trained to learn abstract concepts such as "tree" or "car". Particularly, for the analysis of simple, RNAscope derived fluorescence signals (i.e., individual small dots, diameter Ø ~ 0.3 µm), we found that such a complicated pre-trained neural network was not of benefit. RNAscope-derived fluorescence signals are mainly small dots (Ø ~ 0.3 µm) within the cytoplasm, which makes threshold based automatic recognition/segmentation of the cell body difficult. Thus, we custom-designed and optimized two less complex FCN-based algorithms (DLAP-1 and 2), and we demonstrate that they were better suited for the analysis of RNAscope data, compared to DeepLab3 approaches. The latter algorithm failed to recognize the small dots (mRNA molecules) as evident from the sensitivity-value of 0.0% for this class, compared to 92.5% and 94.7% for the FCN-based algorithm (Table S1). For protein-derived immuno-histochemistry signals (diaminobenzidine (DAB), and

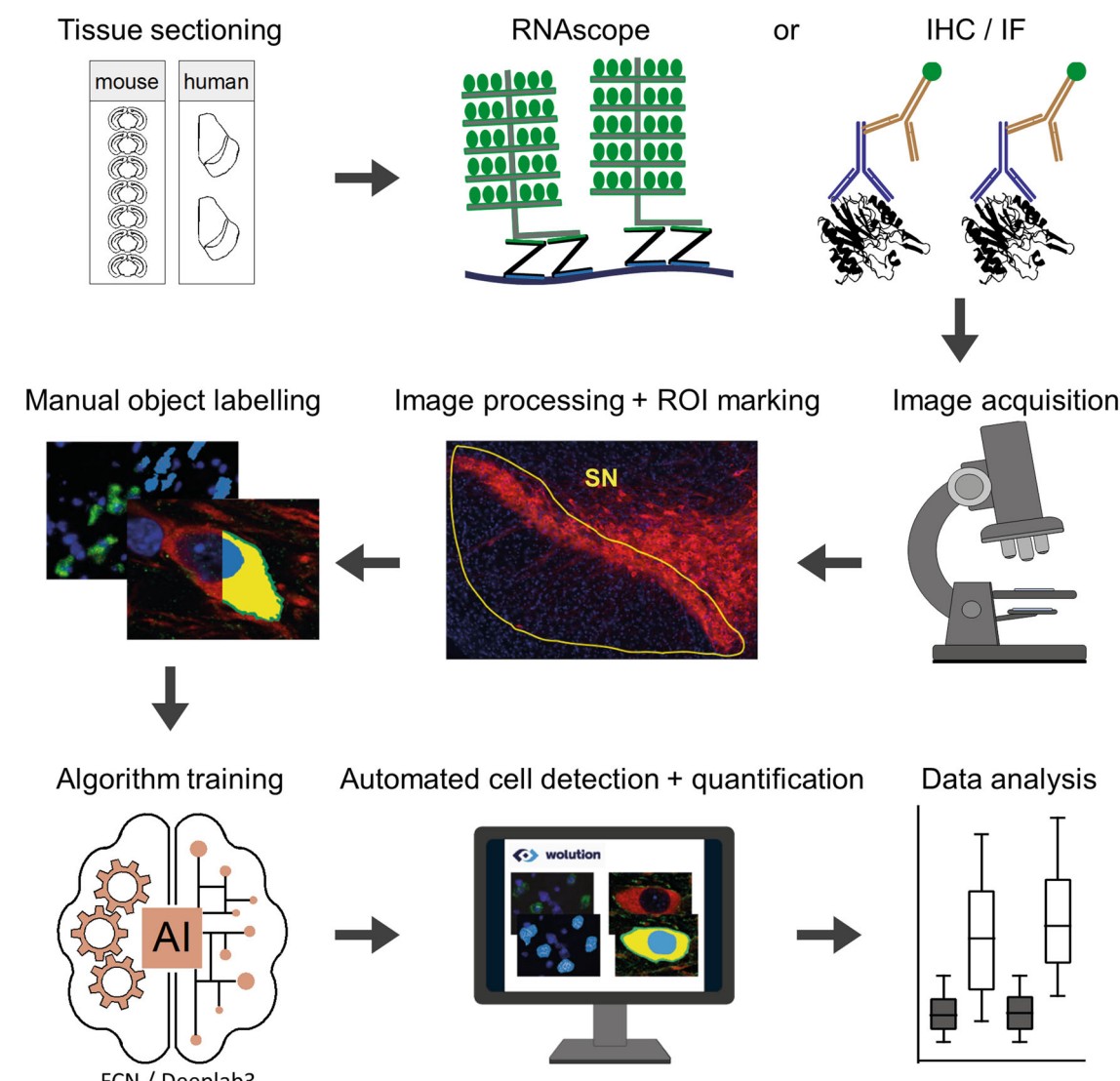

**Fig. 1 Graphic illustration of the deep learning-based image analysis platform (DLAP).** After sectioning of murine/human midbrain samples, either mRNA- (RNAscope) or protein- (IHC/IF) labelling is performed. After image acquisition, images are preprocessed and the ROIs (e.g., SN) are manually marked. Cell types and compartments (e.g., cell body, plasma membrane, nucleus) were manually labelled in a training dataset to generate the ground truth data that was used to train the individual algorithms (FCN- or Deeplab3-based, depending on complexity of signals to identify). After sufficient training, images are automatically analysed (detection of ROI, quantification of cell numbers and signal intensities).

**Table 1 Quality evaluation parameters for all six algorithms.**

| DLAP algorithm | class name | Error [%] | TP [%] | TN [%] | FP [%] | FN [%] | Spec [%] | Sens [%] |
|---|---|---|---|---|---|---|---|---|
| 1 | cell body | 5.08 | 10.3 | 84.7 | 4.1 | 0.8 | 95.3 | 92.7 |
|  | mRNA |  | 0.3 | 98.9 | 0.7 | 0.0 | 99.3 | 92.5 |
| 2 | cell body | 4.13 | 5.2 | 91.0 | 2.5 | 1.4 | 97.3 | 80.8 |
| (human) | mRNA |  | 0.2 | 98.9 | 0.9 | 0.0 | 99.1 | 94.7 |
| 3 | nucleus | 0.61 | 0.5 | 98.9 | 0.5 | 0.1 | 99.5 | 85.5 |
| 4 | nucleus | 0.26 | 0.1 | 99.7 | 0.2 | 0.0 | 99.8 | 59.8 |
| 5 | nucleus | 3.3 | 0.9 | 98.5 | 0.3 | 0.3 | 99.7 | 74.5 |
|  | cytoplasm |  | 4.1 | 92.9 | 2.2 | 0.8 | 97.7 | 83.7 |
| 6 | nucleus | 5.49 | 3.0 | 93.5 | 2.5 | 1.0 | 97.4 | 75.6 |
|  | cytoplasm |  | 9.7 | 89.0 | 0.8 | 0.5 | 99.1 | 95.0 |
|  | membrane |  | 11.7 | 85.2 | 2.2 | 0.9 | 97.5 | 92.4 |

Performances were determined by comparing DLAP analysis results of the test image sets to the ground truth data (provided by manual labelling). Class name defines the identified molecular or cellular structures. The following quality measures were assessed separately for each class: pixel error rate, rate for true positive (TP), true negative (TN), false positive (FP), false negative (FN) detection, as well as relative specificity (spec.) [defined as TN/(TN + FP)], and sensitivity (sens.) [defined as TP/(TP + FN)].

**Table 2 Overview of six individual DLAP algorithms and workflow, optimized for distinct image sets and analysis tasks.**

| DLAP algorithm | Trained by /optimized for | Algorithm task | Network-Type | Training-set | | post training image analysis optimization (a-e) |
|---|---|---|---|---|---|---|
| | | | | images | cells | |
| 1 | RNAscope fluorescence: TH-mouse (DAPI) | (1) Identifies cells of interest and delineates the shape of cell bodies by TH-derived fluorescence signals (green channel). (2) Within these marked cell bodies, target gene derived individual fluorescent dots (red channel) are detected, marked, and counted. | FCN | 100 | ~1000 | (a) watershed algorithm for improved separation of mRNA molecules. (b) morphological operations (type dilation and erosion) applied, for better separation of neurons. |
| 2 modification of algorithm No. 1 | RNAscope fluorescence: TH-human (DAPI) | (3) A fourth BF channel was included, as a measure for NM-content. After Step (1) and (2) from algorithm 1, in an additional step, the BF signal intensity is quantified within the marked cell bodies (including nuclei). | FCN | 200 | ~500[#] | (a) watershed algorithm for improved separation of mRNA molecules. (b) morphological operations (type dilation & erosion) applied, for better separation of neurons. |
| 3 | DAB-labelling: TH-mouse | Identifies and counts nuclei of DAB-positive cells (TH-stained). | DeepLab3[#] | 100 | ~3000[#] | none |
| 4 modification of algorithm No. 3 | DAB-labelling: TH-mouse & HE-counterstain | Extends the automated counting of DAB-positive cells (TH-stained) to sections that are counterstained with HE. | DeepLab3[#] | 300 | ~2500[#] | (a) watershed algorithm for improved separation of individual nuclei. (c) filling holes within recognized compartments. |
| 5 | IF-labelling: TH-mouse, DAPI DAT-mouse D2/CB/Aldh1A1-mouse | (1) Identifies and counts cells of interest (i.e TH-positive cells). It delineates the shape of the cell body by immunofluorescence signals for TH (red channel) and of the nucleus by DAPI fluorescence signal (blue channel). (2) Within these marked compartments, the relative signal intensities are quantified for all four fluorescent channels. | DeepLab3 | 95 | ~500 | (a) watershed algorithm for improved separation of neurons. (c) filling holes within recognized compartments. (d) removing target gene channel during cell identification. (e) RGB intensity threshold for background-signal determination. |
| 6 | IF-labelling: TH-mouse, DAPI Kv4.3-mouse, | (1) Identifies cells of interest and delineates the shape of the cell body by immunofluorescence signals for TH (red channel), of the nucleus by DAPI fluorescence signal (blue channel), and of the membrane (originally learned by Kv4.3 staining, after training no membrane marker needed). (2) Within these marked compartments, the relative signal intensities are quantified for all three fluorescent channels. | DeepLab3 | 94 | ~110 | (c) filling holes within recognized compartments. (d) removing target gene channel during cell identification. (e) RGB intensity threshold for background-signal determination. |

For detailed description, see methods.
BF brightfield, DAB diaminobenzidine, DAPI 4′,6-diamidino-2-phenylindole, FCN fully convolutional neural network, HE hematoxylin, IF immunofluorescence, NM neuromelanin, RGB red-green-blue, TH tyrosine hydroxylase.
[#]For training, only the cellular nuclei were marked, not the full cell body, similar as in stereology; only clearly visible, full nuclei were counted.

immunofluorescence (IF)), tailored DeepLab3 algorithms performed better, as indicated by lower overall error (Table S1; DLAP-3 to -6).

**DLAP based automated quantification of mRNA molecules in individual target cells via RNAscope approaches.** RNAscope in situ hybridization allows absolute quantification of distinct mRNA molecules, directly via hybridization-probes, with each fluorescence dot representing one target mRNA-molecule[52,53]. Due to multiple probes for each target-gene, it is hardly affected by degraded RNA. We have used RNAscope for determining mRNA molecule numbers in individual TH-positive SN DA neurons in fresh-frozen, PFA-postfixed mouse midbrain sections[54]. For automated quantification of numbers of RNAscope probe derived fluorescence dots, two custom-designed FCN based algorithms were used (DLAP-1 & 2). The general principle of the FCN algorithms is given in Fig. 2. DLAP analysis of RNAscope signals was performed in two steps: in a first step, the cell body was recognized and marked

according to the TH-signals, and in a second step, individual target gene mRNA dots (labelled with a different fluorophore) were identified and counted within the marked TH-positive cell body, including the nucleus. DLAP-1 is optimized for detecting TH-positive DA neurons from mice, DLAP-2 for detecting human TH-positive DA neurons, containing neuromelanin (NM), a dark pigment present in DA neurons from humans and primates[55,56], visible only in bright field (BF).

Figure 3a shows a representative image of TH-positive SN neurons in a coronal mouse brain section, after RNAscope in situ hybridization, before (input image, left) and after automated identification of TH-positive cell bodies and quantification of individual fluorescence dots, derived from target mRNA molecules. Cell bodies were identified and marked according to the TH-derived fluorescence signal (green dots), and target mRNA were identified as probe-derived fluorescence-signals (red dots, Table S2). Results for two different target genes - with lower (Cav1.3) and higher (Cav2.3) mRNA abundance - are given (Fig. 3b and Tables S3, S4). Both

target genes code for a specific voltage gated $Ca^{2+}$ channel α-subunit that has been linked to Parkinson's disease[54,57,58]. The manual and the automated analysis (DLAP-1) resulted in very similar results for both genes, over the full range of mRNA molecule numbers (Cav1.3 manual: $24 \pm 1$ mRNA molecules, DLAP-1: $24 \pm 1$, $N = 3$, $n = 499$ neurons analysed, $p = 0.999$, $R^2 = 0.92$; Cav2.3 manual: $69 \pm 5$, DLAP-1: $69 \pm 5$; $N = 3$, $n = 372$; $p = 0.999$, $R^2 = 0.92$; Fig. 3b/c, S1, Tables S3, S4). Accordingly, the significant, ~3-fold higher number of Cav2.3 mRNA molecules in mouse SN DA neurons, compared to those of Cav1.3 is robustly detected with both manual and DLAP-1 analysis ($p < 0.0001$ for both), similar as previously described[54].

Similar robust results were obtained for RNAscope analysis of human SN DA neurons with the DLAP-2 algorithm. To optimize DLAP for human SN DA neurons, the DLAP-1 algorithm was extended to quantify the brightfield (BF) intensity, as a measure for the variable NM content of human SN DA neurons, to evaluate possible confounding effects of different NM content on RNAscope results (Fig. 4, S2, Table 2). Cav1.3 mRNA molecules as well as NM content were compared in SN DA neurons from adult and aged individuals (adult: $42 \pm 7$ years; aged: $78 \pm 3$, details in Table S5). Manual and automated DLAP-2 analysis provided very similar results, and we detected a strong linear correlation between manually and automatically determined mRNA molecule numbers as well as NM-content (Fig. 4b/d/e, Tables S6, S7). Interestingly, the number of Cav1.3 mRNA molecules was significantly lower (~60%) in SN DA neurons from aged individuals (Fig. 4b left, manual: adult $38 \pm 5$, aged $14 \pm 3$; $N = 3$, $n = 223$; $p < 0.0001$; DLAP-2: adult $40 \pm 7$, aged $17 \pm 4$; $N = 3$, $n = 258$; $p < 0.0001$).

Noteworthy, the BF-value derived NM content was also significantly different between adult and aged samples, with ~30% higher NM content in aged SN DA neurons ($p < 0.0001$, Fig. 4b right, Tables S6, S7). To exclude that the lower number of mRNA molecules determined in aged SN DA neurons were artificially caused by the higher BF values (as measure for NM content), we addressed possible confounding parameters by applying a mixed effects model optimized for these kind of human sample-derived data that are in general more heterogeneous[59,60]. As the model indicated that the NM content indeed affected the detected Cav1.3 mRNA molecule numbers, we corrected the automated data for the NM-content attributed effect. However, as evident in Fig. 4c, Table S6, we still detected a similar highly significant lower number of Cav1.3 mRNA molecules (~35%) in SN DA neurons from aged donors (manual: adult $32 \pm 2$, aged $17 \pm 3$; $N = 3$, $n = 180$; $p < 0.0001$; DLAP-2: adult $32 \pm 3$, aged $20 \pm 3$; $N = 3$, $n = 244$; $p < 0.0001$).

**DLAP based automated quantification of immuno-labelled neuron numbers in tissue sections.** Determination of numbers of cellular populations in defined ROI is a common task e.g., for quantifying selective cell loss in neurodegenerative diseases, and to assess neuroprotective therapeutic effects[61]. However, commonly

used stereological approaches (still the gold-standard), are time consuming, and they provide only a - more or less exact - estimation of cell numbers, depending on the size of the manually counted fraction of cells[62,63]. Approaches have been developed to increase sampling sizes and thus accuracy of estimates[64]. Nevertheless, estimates are prone to over- or underestimating total cell numbers, particularly if the target cells are not homogeneously distributed in the ROI, as it is the case e.g., due to differential neuronal vulnerability in degenerative disease[65-67]. However, manual approaches to count all neurons in a given ROI are extremely time consuming and prone to bias/human error. To overcome these issues, we developed three DLAP algorithms, dramatically reducing the time for valid determination of cell numbers, tailored to detect and count target cells, immuno-labelled by either DAB alone (DLAP-3), or in the presence of an additional hematoxylin counterstain (DLAP-4), or by IF (DLAP-5). According to our stereological procedures, the DLAP-3 and 4 algorithms were trained by marking the neuronal nuclei only, not the whole cell bodies, to count only TH-positive cells with clearly visible nuclei.

DAB-labelling is commonly used for determining cell-numbers in histological samples, as stained sections can be archived and e.g. (re-) analysed. Figure 5, S3, Tables S8, S11 summarize the results for counts of TH-positive SN neurons (unilaterally) from juvenile mice after DAB immunostaining, comparing stereological estimates with automated counting, utilising DLAP-3. The stereological estimates and automated neuron counts resulted in very similar neuron numbers (Stereology: $4951 \pm 1079$, DLAP-3: $4317 \pm 912$ neurons unilaterally; $p = 0.218$, $N = 10$. linear correlation/section: $R^2 = 0.79$). These SN DA neuron numbers correspond well to the literature for C57bl/6 mice, ranging from 4000–6000 neurons unilaterally[68,69].

Immunohistochemistry sections are often counterstained with an unspecific cellular marker (like Nissl, hematoxylin). For analysis of SN DA neurons, such counterstains are often performed to confirm that a loss of TH-immunopositive neurons (e.g., in a PD-model) indeed corresponds to a loss of the respective neurons, and does not reflect a mere downregulation of TH-expression[70-72]. As we use hematoxylin-counterstains in such experiments[54,73], we optimized DLAP-4 to detect DAB stained TH-positive SN neurons in such counterstained sections. With DLAP-4, we re-analysed a cohort of 21 mice that we had already analysed[54] to quantify the effect of a PD-inducing drug (1-Methyl-4-phenyl-1,2,3,6-tetrahydropyridin, MPTP) that induces preferential loss of SN DA neurons. Adult C57bl/6 J mice were either injected with saline ($N = 9$) or with MPTP/probenecid ($N = 12$). We compared DLAP-4 analysed data with the respective already published stereology-derived data, and also with another automated neuron count approach, based on the Aiforia platform (Fig. 6, S4, Tables S9–11). We had used the Aiforia platform (https://www.aiforia.com) for this cohort, to confirm relative numbers of remaining SN DA neurons after drug-treatment. Indeed, the relative determined cell loss by the Aiforia approach was similar to that estimated by stereology (~40%, $p = 0.183$; Fig. 6b, Table S10,

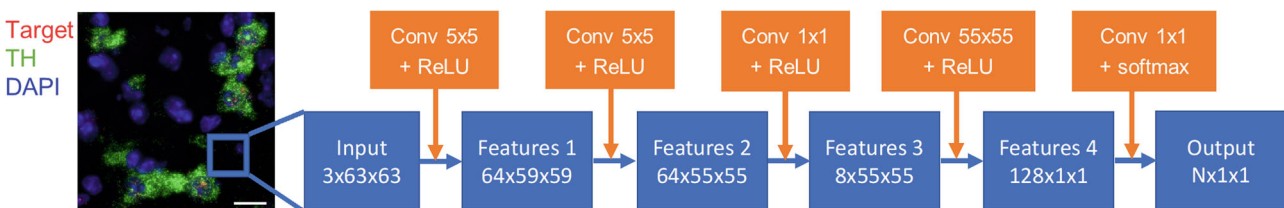

**Fig. 2 Architecture of the fully convolutional neural network (FCN), used for RNAscope-derived signals.** Left: the input image shows SN DA neurons, RNAscope-labelled for tyrosine hydroxylase (TH) mRNA (green), target gene mRNA (red) and DAPI (blue). Scale bar: 20 μm. Right: the image is passed through five convolutional layers, connected by the Rectified Linear Units (ReLU) and the softmax activation function, as indicated. The latter normalises the network output to the probability distribution. The size of the convolutional kernels is specified in the orange boxes, the dimensions of the resulting feature maps are given in the blue boxes [pixels]. The network analyses image patches of size 63 × 63 pixels to determine the class of the central pixel in the patch. It is applied convolutionally to every central pixel in the image, so that the final output image has the same dimensions as the input image.

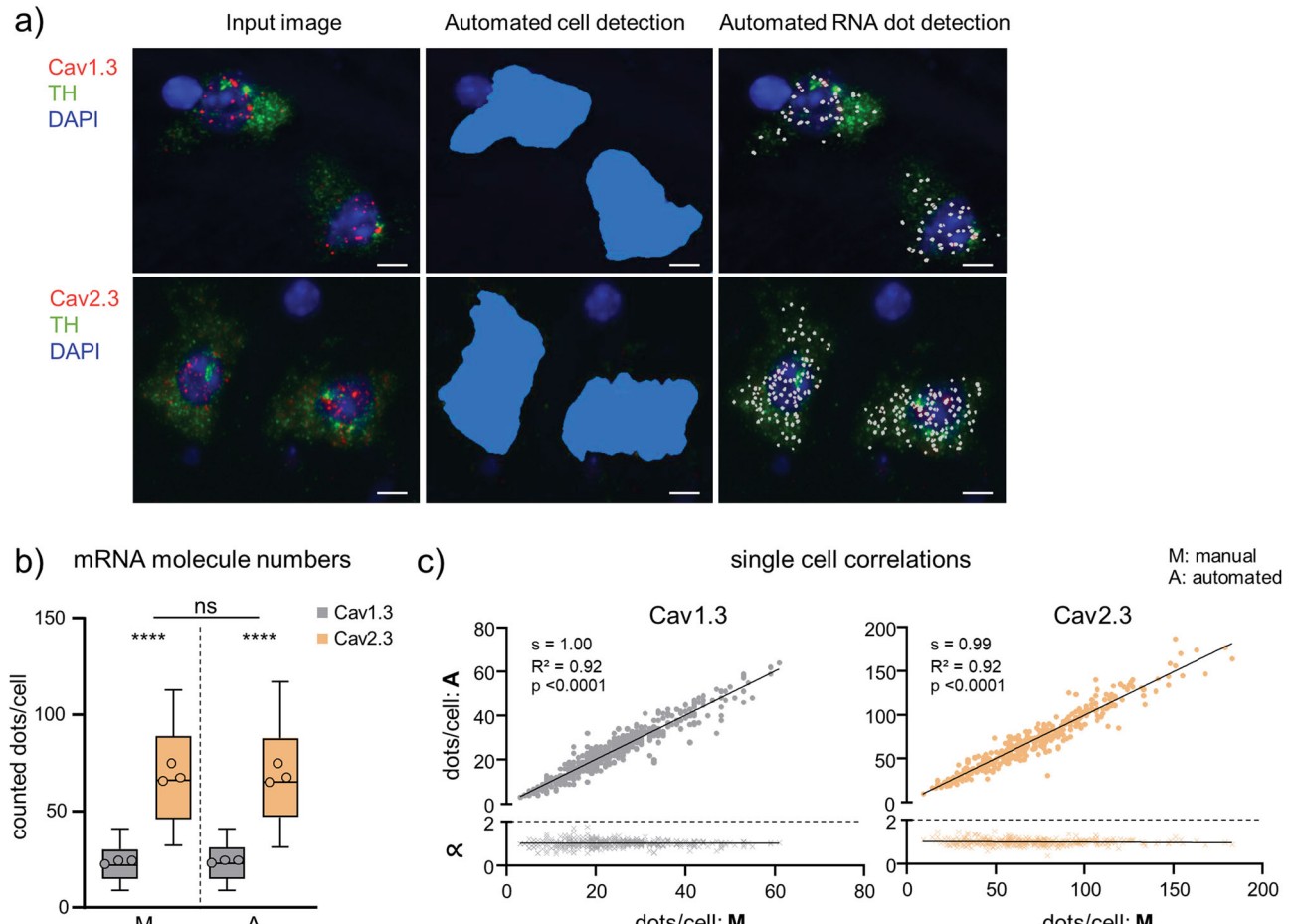

**Fig. 3 Comparison between manual and automated (DLAP-1) RNAscope-derived image analysis of mouse midbrain sections. a** SN DA neurons in coronal midbrain section from adult mice after RNAscope for tyrosine hydroxylase (TH, green) and for the voltage gated $Ca^{2+}$ channel α-subunits Cav1.3 or Cav2.3 (red). Nuclei are marked by DAPI-staining (blue). Left: input images. Middle and right: images after automated recognition of the TH-labelled cell bodies (middle, blue area) and of target mRNA molecules (right, white dots) by DLAP-1. Scale bars: 5 μm. **b** mRNA molecule numbers/neuron, counted manually (M) or automatically (A) via DLAP-1, as indicated. Data are given as boxplots (median, 10–90 percentile) for all analysed neurons (Cav1.3: $n = 449$, Cav2.3: $n = 372$). The open circles indicate mean values for each analysed animal ($N = 3$ each). Significant differences (for analysed neurons, n) according to two-way ANOVA with Tukey's multiple comparison tests (****$p < 0.0001$, ns: $p > 0.05$). **c** Upper: single neuron correlations of mRNA-derived dot counts/cell between manual and automated analysis according to Pearson correlation test (s = slope). Lower: corresponding proportionality constants ∝, calculated from the manual and automated dot count ratios. All data are detailed in Tables S3, S4 and Fig. S1.

and Benkert, Hess[54]). However, the absolute number of neurons determined with the Aiforia approach was almost double as high (saline: stereology: $3959 \pm 643$, Aiforia: $7048 \pm 843$, $p < 0.0001$, drug: stereology: $2504 \pm 933$, Aiforia: $4108 \pm 1596$, $p = 0.004$; Fig. 6b, S4, Table S9). This overcount, and the fact that the convolutional neuronal network (CNN)-based Aiforia system does not allow algorithm training by the users[74], prompted us to switch to the Wolution-platform in the first instance.

With the DLAP-4 approach, a similar relative % of remaining TH-positive SN neurons in the drug-treated mouse cohort was detected, as with stereology or Aiforia (DLAP-4: $59 \pm 21\%$, stereology: $63 \pm 59\%$, $p = 0.234$; Aiforia: $58 \pm 23\%$, $p = 0.988$). Moreover, also the absolute numbers of SN DA neurons/mouse, counted by DLAP-4 were similar to the respective stereological estimates (Saline: DLAP-4: $4614 \pm 500$, stereology: $3959 \pm 643$, $p = 0.744$; Aiforia: $7048 \pm 823$, $p < 0.0001$. Drug: DLAP-4: $2708 \pm 975$, stereology: $2504 \pm 922$, $p = 0.996$; Aiforia: $4108 \pm 1596$, $p = 0.016$). The linear correlation coefficients between manually and automatically counted neuron numbers were similarly high for both automated approaches. However, the linear regression slope, was higher for the Aiforia algorithm

compared to our DLAP-4 algorithm (1.6 vs 1.0), in line with the higher, over-counted neuron-number by the Aiforia approach (Fig. 6, S4, Table S11).

**DLAP-5 analysis of immunofluorescence-labelled cells defines a novel population of DAT-negative neurons in the lateral SN.** We extended our automated cell quantification approach from DAB/BF to cells labelled by immunofluorescence (IF). IF has a greater dynamic signal range[75,76], and allows the parallel quantification of more than one cell-population, defined by distinct markers[77]. Moreover, it can provide additional information regarding protein localization and relative expression levels.

We trained DLAP-5, based on TH and DAPI signals only, to detect and count immunofluorescence TH-positive SN DA neurons in DAPI co-stained sections, and to mark the neuronal cell body and the nucleus of TH-positive neurons. After identification of TH-positive cell bodies and nuclei, in the next step, DLAP-5 quantifies IF-intensities separately in both identified compartments for currently up to four distinct fluorescent channels, enabling co-expression analyses. Fig. 7a

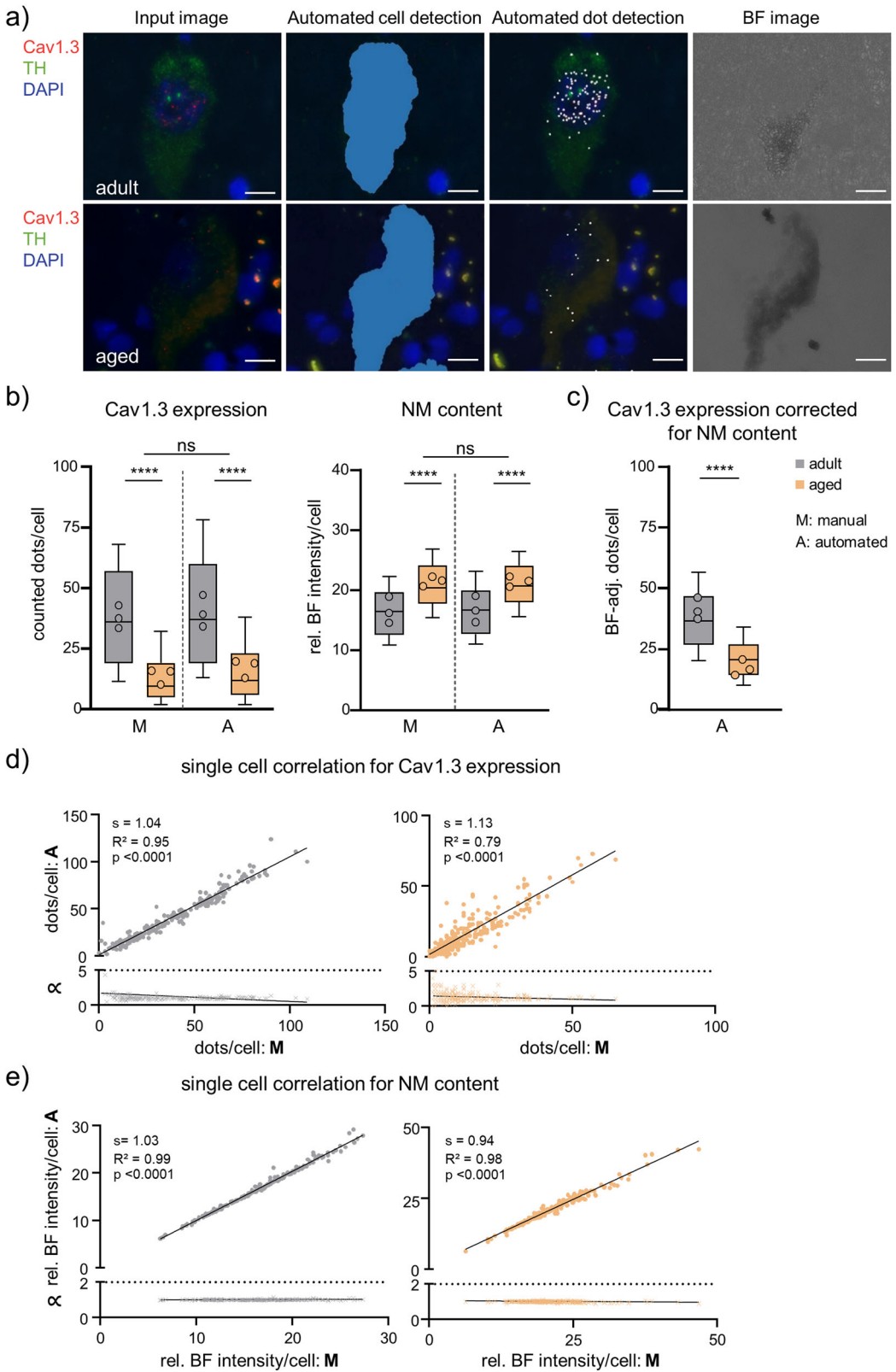

and S5a illustrate the automated identification of TH IF-positive SN neurons (red) in a coronal mouse midbrain section.

As the expression of the dopamine transporter DAT is commonly used as an additional maker for SN DA neurons besides TH, we used DLAP-5 to quantify the relative DAT-IF signals in the marked TH-positive cell bodies. Our central aim here was to quantify the co-expression of TH and DAT in SN neurons, and their relative expression levels, rather than determining absolute numbers of TH-positive SN neurons (Figs. 7–9, S5–S7). Thus, we only included those TH-positive SN neurons into further quantitative DAT analysis that were clearly separated/segmented, with marked cell bodies of proper size and clearly visible full nuclei. Our exclusion criteria are specified in methods and in Fig. S5b, and resulted in manual

**Fig. 4 Comparison between manual and automated (DLAP-2) RNAscope-derived image analysis of human midbrain sections. a** SN DA neurons in a horizontal midbrain section from human adult and aged *post mortem* brain samples after RNAscope for tyrosine hydroxylase (TH, green) and for the voltage gated $Ca^{2+}$ channel α-subunit Cav1.3 (red). Nuclei are marked by DAPI-staining (blue). Left: input images. Middle: images after automated recognition of the TH-labelled cell bodies (middle left, blue area) and of target mRNA molecules (middle right, white dots) with DLAP-2. Right: corresponding brightfield (BF) images used to quantify neuromelanin (NM) content. Relative (rel.) BF intensities were measured within the detected cell bodies. Scale bars: 10 μm. **b** mRNA molecule numbers/neuron (left) and NM content (rel. BF signal intensity; right), quantified manually (M) or automatically (A) via DLAP-2, as indicated. Data are given as boxplots (median, 10–90 percentile) for all analysed neurons (adult: *n* = 223, aged: *n* = 258). The open circles indicate mean values for each analysed brain sample (N = 3 each). Significant differences (for analysed neurons, *n*) according to two-way ANOVA with Tukey's multiple comparison tests (****: *p* < 0.0001, ns: *p* > 0.05). **c** mRNA molecule numbers/SN DA neuron, determined via DLAP-2, adjusted for individual NM using a Mixed Effects Model. Data are given as boxplots (median, 10–90 percentile) for all analysed neurons (adult: *n* = 206, aged: *n* = 231). The open circles indicate mean values for each analysed brain sample (N = 3 each). Significant differences (for analysed neurons, *n*) according to Mann–Whitney tests (****p* < 0.0001). d/e) Upper: single neuron correlations of Cav1.3 mRNA-derived dot counts/cell (**d**) and rel. BF signal intensity/cell (**e**) between manual and automated analysis according to Pearson correlation test. Lower: corresponding proportionality constants ∝, calculated from the manual and automated dot count ratios. All data are detailed in Tables S5–7 and Fig. S2.

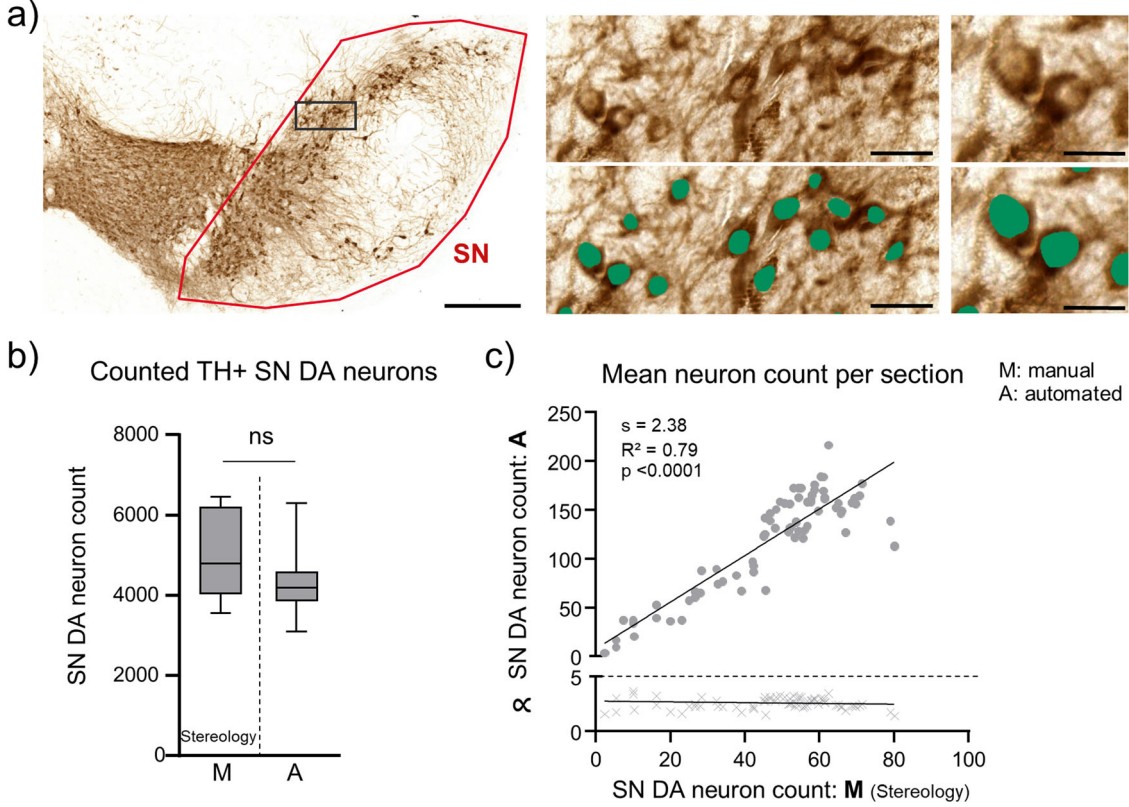

**Fig. 5 Comparison between unbiased stereology and automated (DLAP-3) IHC-derived image analysis of juvenile mouse midbrain sections. a** SN DA neurons in coronal midbrain section from juvenile mice after DAB-IHC for tyrosine hydroxylase (TH, brown). Left: input image with SN outlined in red. Scale bar: 200 μm. Middle and right: enlarged images before (top) and after (bottom) automated recognition of the nuclei within TH-labelled cell bodies (green area) with DLAP-3. Scale bars: 50 μm (middle) and 10 μm (right). **b** Numbers of TH-positive SN DA neurons/mouse, quantified via unbiased stereology (extrapolated via the optical fractionator method, M) or automatically (A) via DLAP-3, as indicated. Data are given as boxplots (median, 10–90 percentile) for all analysed mice (N = 10). No significant differences according to Mann–Whitney test (ns: *p* > 0.05). **c** Upper: correlations of SN DA neuron counts/section between stereology and automated analysis, according to Pearson correlation test. Lower: corresponding proportionality constants ∝, calculated from the manual and automated cell count ratios. All data are detailed in Tables S8, S11 and Fig. S3.

exclusion of ~30 ± 6% of the automatically identified TH-positive SN neurons (Fig. S5c, Table S12). In the remaining number of TH-positive SN neurons (2750 ± 462 neurons/mouse, N = 14), we detected only in ~95% a DAT-IF signal above the background signal (2633 ± 442 neurons/mouse, *p* = 0.0001). The remaining ~5% (4.3 ± 1.7%) of TH-positive SN neurons were immunofluorescence negative for DAT (Fig. 7b, S5d, Tables S12, S13, S16).

In order to further characterize this small subpopulation of TH-positive but DAT-negative SN neurons, we used the x,y,z

-coordinates of the analysed TH-positive neurons to generate 2D and 3D anatomical maps for each analysed coronal section, and merged maps for all mice Figs. 7c, 8, S6, online-Fig. S7, Table S16). These maps revealed that the TH-positive DAT-negative neurons were clustered in the very lateral parts of the SN, with 17% of all TH-positive neurons being DAT-negative compared to only ~3% in the whole non-lateral SN (lateral – defined as >1.5 scaled x-axis units (>377.8 μm) lateral from each SN hemisphere-centre (0.0): DAT negative: 17.0 ± 5.8%, non-lateral: 3.2 ± 1.6%, Fig. 7d, S6). This location excludes that they constitute a contamination

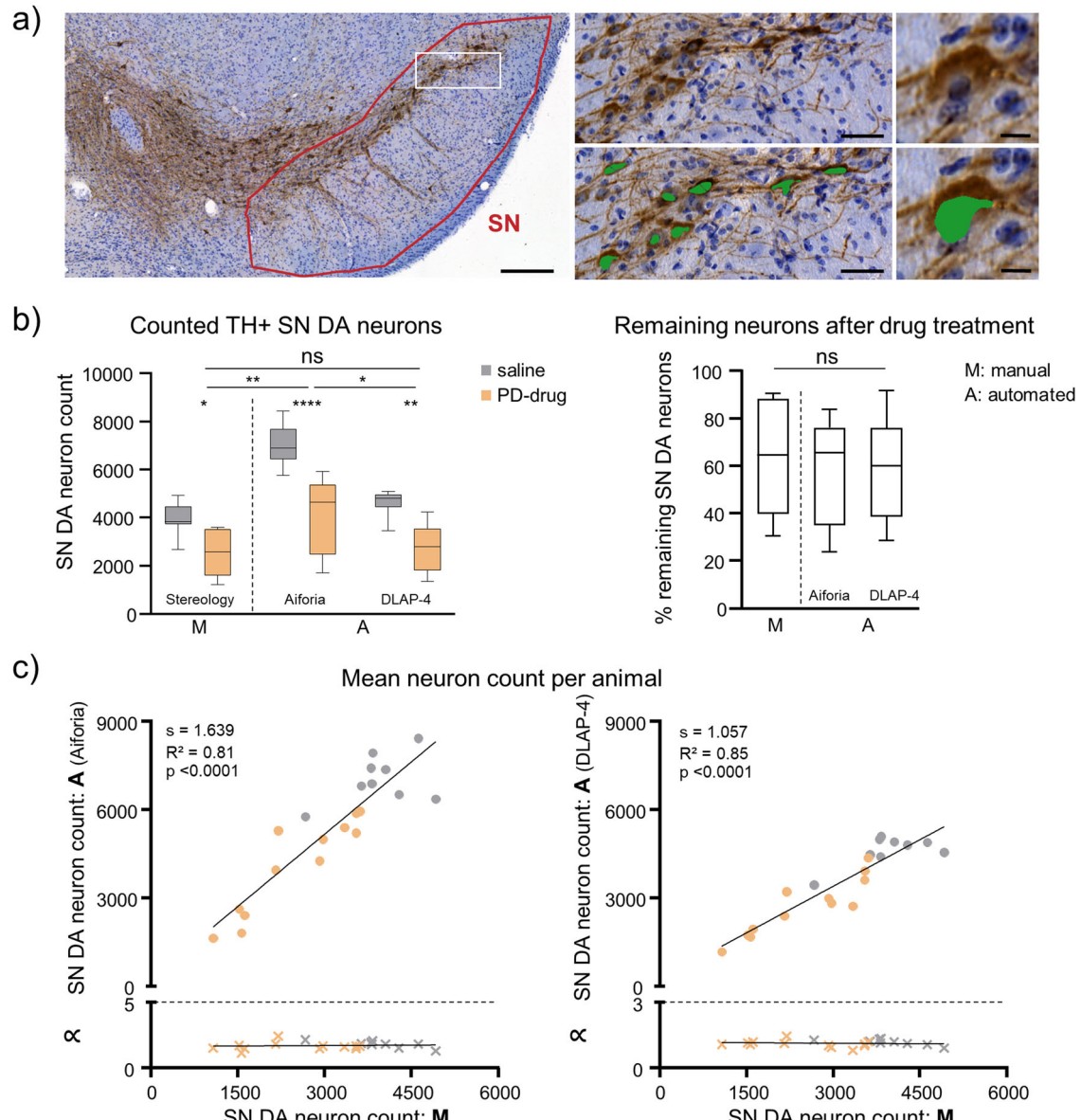

**Fig. 6 Comparison between unbiased stereology and automated (DLAP-4) IHC-derived image analysis of adult mouse midbrain sections. a** SN DA neurons in coronal midbrain section from adult mice after DAB-IHC for tyrosine hydroxylase (TH, brown). Nuclei are marked by hematoxylin-staining (blue). Left: input image where the SN is outlined in red. Scale bar: 200 μm. Middle and right: enlarged images before (upper) and after (lower) automated recognition of the nuclei within TH-labelled cell bodies (green area) with DLAP-4. Scale bars: 50 μm (middle) and 10 μm (right). **b** Left: number of TH-positive SN DA neurons/mouse quantified via unbiased stereology extrapolated via the optical fractionator method (M), or automatically (A), either via an algorithm provided by Aiforia or DLAP-4, as indicated. Right: remaining TH-positive neurons [%] in mice, treated with a neurodegenerative drug, calculated relative to the mean of the saline-treated group. Data are given as boxplots (median, 10–90 percentile) for all analysed mice (saline: $N = 9$; PD-drug: $N = 12$). Significant differences, according to two-way ANOVA with Tukey's (neuron counts) or Kruskal–Wallis with Dunn's multiple comparison tests (percentage of remaining neurons) (ns: $p > 0.05$, *$p < 0.05$, **$p < 0.01$, ****$p < 0.0001$). **c** Upper: correlations of SN DA neuron counts/animal between stereology and automated analysis (left: Aiforia, right: DLAP-4), according to Pearson correlation test. Lower: corresponding proportionality constants ∝, calculated from the manual and automated cell count ratios. Stereology data and Aiforia data modified from Benkert et al., Nat. Commun. 2019. All data are detailed in Tables S9–11 and Fig. S4.

by VTA DA neurons, overlapping with the medial SN. Moreover, the 3D-maps identified a ~7-fold enrichment of the DAT-negative TH-positive neurons in the caudo-lateral SN, compared to its rostral parts (caudal: $36.7 \pm 14.8\%$, medial: $15.0 \pm 5.3\%$, rostral: $5.0 \pm 5.3\%$, $p < 0.0001$; Fig. 8, S6/7, Table S12), and a ~12-fold enrichment compared to the non-lateral parts of the SN. Moreover, cell bodies of the DAT-negative TH-positive SN neurons were about 35% smaller compared to the DAT-positive neurons (DAT-positive: $195 \pm 15 \, \mu m^2$, DAT-negative:

$131 \pm 17 \, \mu m^2$, $p < 0.0001$), with those of the lateral DAT-negatives being about a further 20% smaller than their non-lateral counterparts, but no further differences between the caudo-lateral and rostro-lateral neurons (DAT-negative: lateral $118 \pm 20 \, \mu m^2$ vs. non-lateral $138 \pm 17 \, \mu m^2$, $p < 0.011$, caudo-lateral $116 \pm 20 \, \mu m^2$ vs rostro-lateral $124 \pm 33 \, \mu m^2$, $p = 0.650$, Fig. 7e, Table S17).

For subpopulations of VTA DA neurons, DAT-expression gradients have been described[78,79]. To systematically address

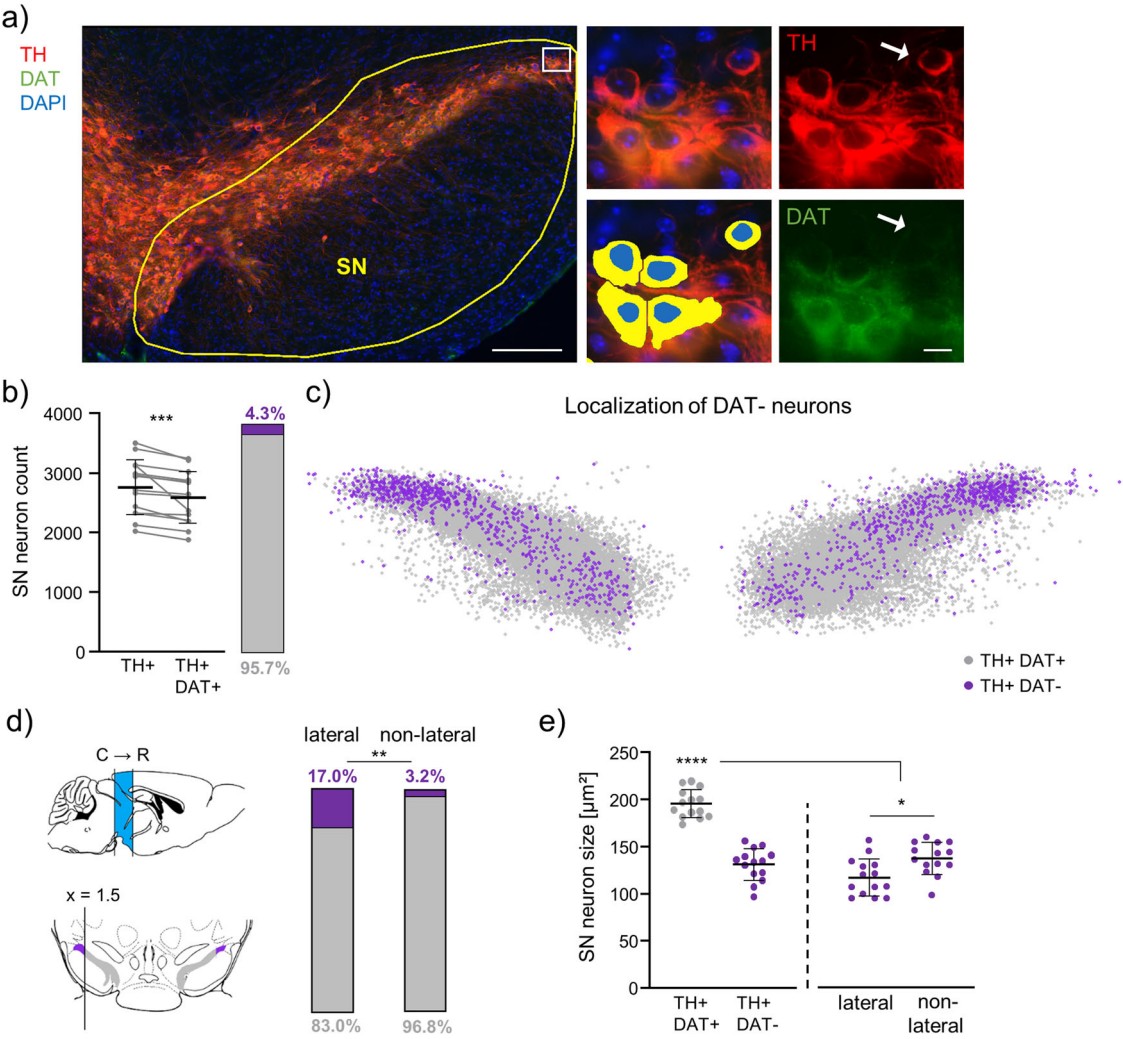

**Fig. 7 Automated (DLAP-5) image analysis of IF-derived signals identifies a DAT-negative subpopulation of TH-positive SN neurons. a** Left: neurons in a coronal midbrain section from adult mice after IF for tyrosine hydroxylase (TH, red) and Dopamine transporter (DAT, green). Nuclei are marked by DAPI-staining (blue), the SN is outlined in yellow, the white box in the lateral SN indicates the location of DAT immuno-negative (DAT−) neurons. Middle: enlarged image of TH-positive neurons within the white box from the left, before (top) and after (bottom) automated recognition of TH-positive cell bodies (yellow) and nuclei (blue) with DLAP-5. Right: single channel input image, for TH-IF (red, top) and DAT-IF (green, bottom). White arrows point to a TH-positive (TH+), DAT-negative (DAT−) SN neuron. Scale bars: 200 μm (left) and 10 μm (middle, right). **b** TH-positive (TH+) and TH- and DAT-positive (TH + DAT + ) neurons/mouse and their relative amounts, quantified via DLAP-5. Given are the numbers of TH+ neurons that were further analysed for semi-quantitative DAT-analysis (compare Fig. S5b for criteria). Data are given as scatterplots and mean ± SD for all analysed mice (N = 14). Significant difference according to Wilcoxon test (***p < 0.001). **c** Plotted are the individual TH+ neurons for all analysed animals (n = 38504, N = 14), according to their scaled x,y-coordinates, determined via DLAP-5. The resulting anatomical 2D-maps display the medio-lateral distribution of the TH + DAT− neurons (violet) within the SN (TH + DAT+ in grey). **d** Left: sagittal and coronal mouse brain sections, modified from (Paxinos & Keith B. J. Franklin, 2007), illustrating the analysed caudo-rostral extent of the SN (bregma: −3.9 to −2.7, sagittal, blue), and the definition of its lateral parts in coronal sections (defined as >1.5 scaled x-units (>377.8 μm) lateral from each SN hemisphere-centre (0.0); lateral: violet, non-lateral: grey, compare Fig. S6). Right: relative amounts of TH + DAT− neurons in the lateral and non-lateral SN. Significance according to Fisher's exact test (**p < 0.01). **e** Mean cell body sizes (x,y-area) of TH + DAT+ and TH + DAT− SN neurons as indicated, determined via DLAP-5. Data are given as scatterplots and mean ± SD for all analysed animals (N = 14). Significant differences according to Mann–Whitney tests (*p < 0.05). All data are detailed in Tables S12, S16, S17, Figs. S5–S7.

whether a gradient in DAT expression is present within the SN, we used the relative, non-saturated DAT fluorescence signal intensities (DAT-RF) of each analysed TH IF-positive neuron, determined by DLAP-5, and the respective x,y,z-coordinates to generate heat-maps of RF-signal intensities (Fig. 9a, S7). Those maps indicated a medio-lateral as well as a caudo-rostral DAT-protein gradient, with lowest DAT-immunosignals (within the background-range) in the very (caudo-) lateral SN. Corresponding heat-maps for TH suggested a similar, but much less prominent gradient (Fig. 9b, S7), similar as previously described[78,80].

To further characterize these atypical TH-positive but DAT-negative lateral SN neurons, we next analysed the co-expression patterns and expression-maps of the dopamine-D2-autoreceptor (D2-AR), the $Ca^{2+}$ binding protein calbindin-d28k (CB), and the aldehyde dehydrogenase 1 (Aldh1A1), all markers for subpopulations of SN DA neurons. D2-AR is known to be abundantly expressed in classical (DAT-positive and CB-negative[81,82]) SN DA neurons[83,84], and accordingly, we robustly detected D2-co-expression in almost all ( ~ 93%) TH- and DAT-positive SN neurons (Figs. 10a, 11a, S8, Tables S14–17). In the DAT-negative SN neurons however, D2-AR co-expression was significantly less

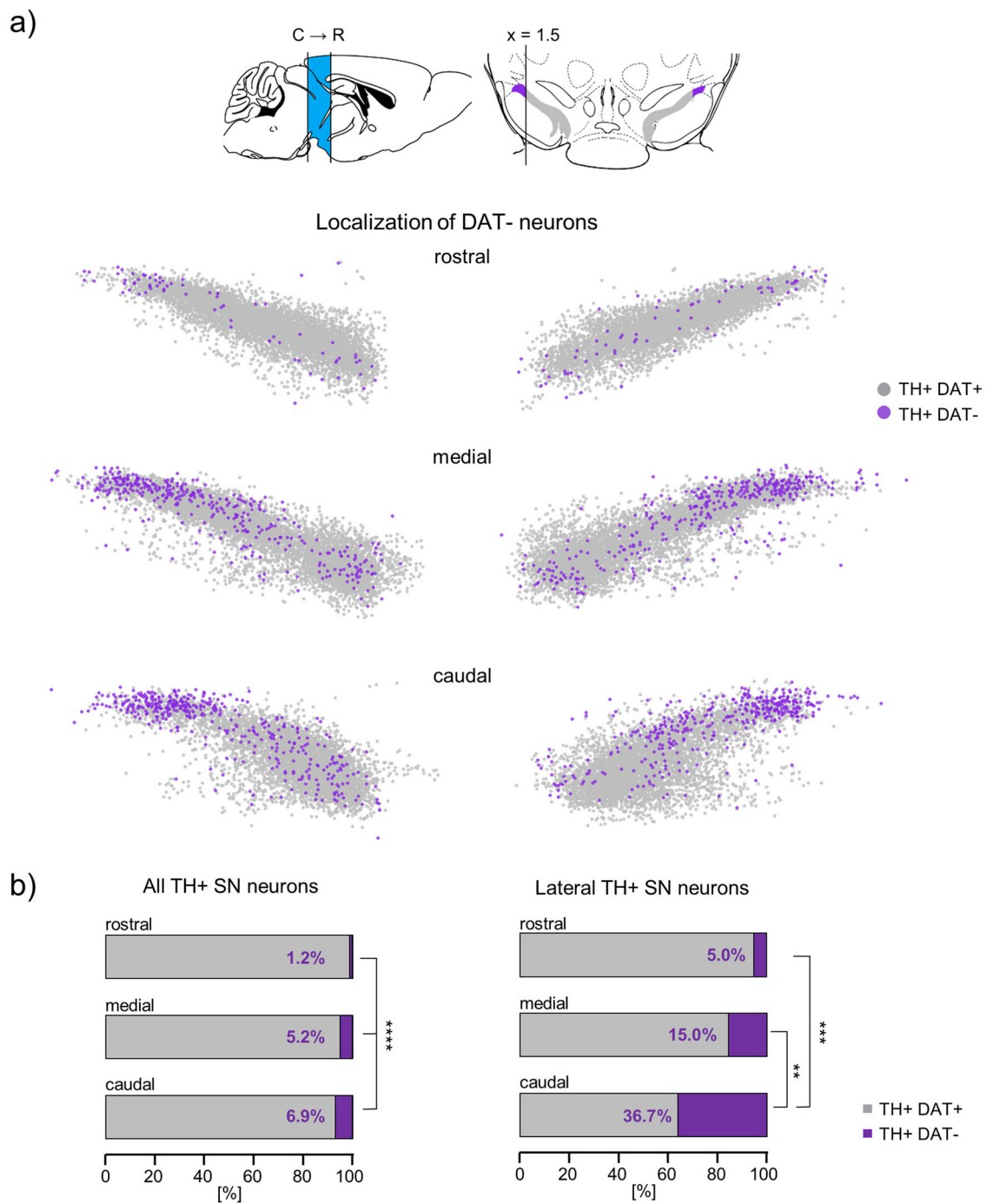

**Fig. 8 Caudo-rostral distribution of TH-positive DAT-negative mouse SN neurons. a** Upper: illustration of the caudo-rostral extent of the SN (bregma: −3.9 to −2.7), the definition of its lateral parts in coronal sections, and colour coding, as in Fig. 7d, S6. Lower: plotted are the individual TH-positive neurons according to their scaled x,y-coordinates, separately for rostral, medial and caudal sections (according to bregma: rostral: −2.7 to −3.1 mm; medial: −3.1 to −3.5; caudal:−3.5 to −3.9; $N = 14$; rostral: TH + DAT− $n = 145$, TH + DAT+ $n = 12263$; medial: TH + DAT− n = 742, TH + DAT+ n = 13821; caudal: TH + DAT− n = 797, TH + DAT+ n = 10736). **b** relative abundancies of TH + DAT+ and TH + DAT− neurons, as indicated. Significant differences according to Chi square tests (**$p < 0.01$, ****$p < 0.0001$). All data are detailed in Table S12. Corresponding 3D maps are given in Fig. S7.

abundant (~70%, $p < 0.0001$; $N = 3$), with no further difference between neurons in the lateral and non-lateral SN. The cytosolic $Ca^{2+}$ binding protein CB is a marker for SN DA neurons that are less vulnerable in PD-paradigms[69,85–87]. Accordingly, and in line with previous publications[78,86,88–90], we detected only in a small percentage of TH- and DAT-positive SN neurons co-expression of CB (9 ± 1%, $n = 865$ from 8446 neurons, $N = 3$); Figs. 10b, 11b,

S8b, Tables S14–17), and the cell bodies of these neurons were ~20% smaller compared to the CB-negative SN DA neurons (TH&DAT + CB + : 173 ± 7 μm², TH&DAT + CB−: 195 ± 8 μm², $p < 0.0001$). In contrast, in the TH-positive but DAT-negative SN neurons, the CB co-expression rate was ~4-fold higher (39 ± 7% $n = 143$ from 383 neurons; $p < 0.0001$, Fig. 11b). Additionally, those neurons had the smallest cell bodies from all

four TH-positive groups; they were only about half of the size of that of classical DAT-positive and CB-negative SN DA neurons (DAT− CB +: $104 \pm 8 \, \mu m^2$; $p < 0.0001$). Moreover, the DAT-negative but CB-positive neurons clustered in the lateral SN, with a ~ 7-fold higher co-expression rate (67%), compared to all DAT-positive SN neurons, (Fig. 11b), had the smallest cell bodies ($96 \pm 5 \, \mu m^2$, Table S17). Among the CB-positive neurons, the relative CB-expression levels were about 12% higher in the DAT-negative SN neurons, compared to the DAT-positives (DAT +: $2.6 \pm 0.9$, DAT−: $3.0 \pm 0.8$, $p < 0.0001$). In contrast, Aldh1A1, a marker for highly vulnerable ventral tier SN DA neurons[87,91] was rather not expressed in the TH-positive but DAT-negative lateral SN neurons (Figs. 10c, 11c, S8c, Tables S14–17), compared to ~60% co-expression in the DAT-positive non-lateral SN DA neurons, with a higher co-expression rate in more medio-ventrally located SN DA neurons, in line with previous descriptions[92,93].

To our knowledge, neurons in the very caudo-lateral SN that are immuno-positive for TH but immuno-negative for DAT and, a medio-lateral and rostro-caudal DAT-gradient in the SN, has not yet been systematically described. We enabled this by analysing DAT-expression in about 40,000 SN neurons - hardly

possible without the automated DLAP-5 approach. However, this algorithm does not allow conclusions regarding the sub-cellular location of the IF-signal.

**DLAP-6 based relative quantification of immuno-fluorescence derived signals in cellular compartments**. As proteins can mediate different functions in dependence of their cellular localization, analysis with higher resolution is desired. To enable relative quantification of immunofluorescence signals in plasma-membranes, cytoplasm and nucleus, we developed the DLAP-6 algorithm, optimized to detect and analyse neurons in high-resolution confocal fluorescent images.

DLAP-6 was specifically trained by using TH as marker for the cell body/cytoplasm, DAPI as marker for the nucleus, and the ion channel subunit Kv4.3 as marker for the plasma-membrane of SN DA neurons. Kv4.3 is not a specific marker for dopaminergic neurons, but it is highly expressed in plasma-membranes of SN DA neurons, and it is the pore-forming subunit of voltage and $Ca^{2+}$ sensitive A-type $K^+$ channels that modulate the activity and the vulnerability of SN DA neurons[94–99]. The $Ca^{2+}$-sensitivity of Kv4.3 channel complexes is mediated by plasma-membrane

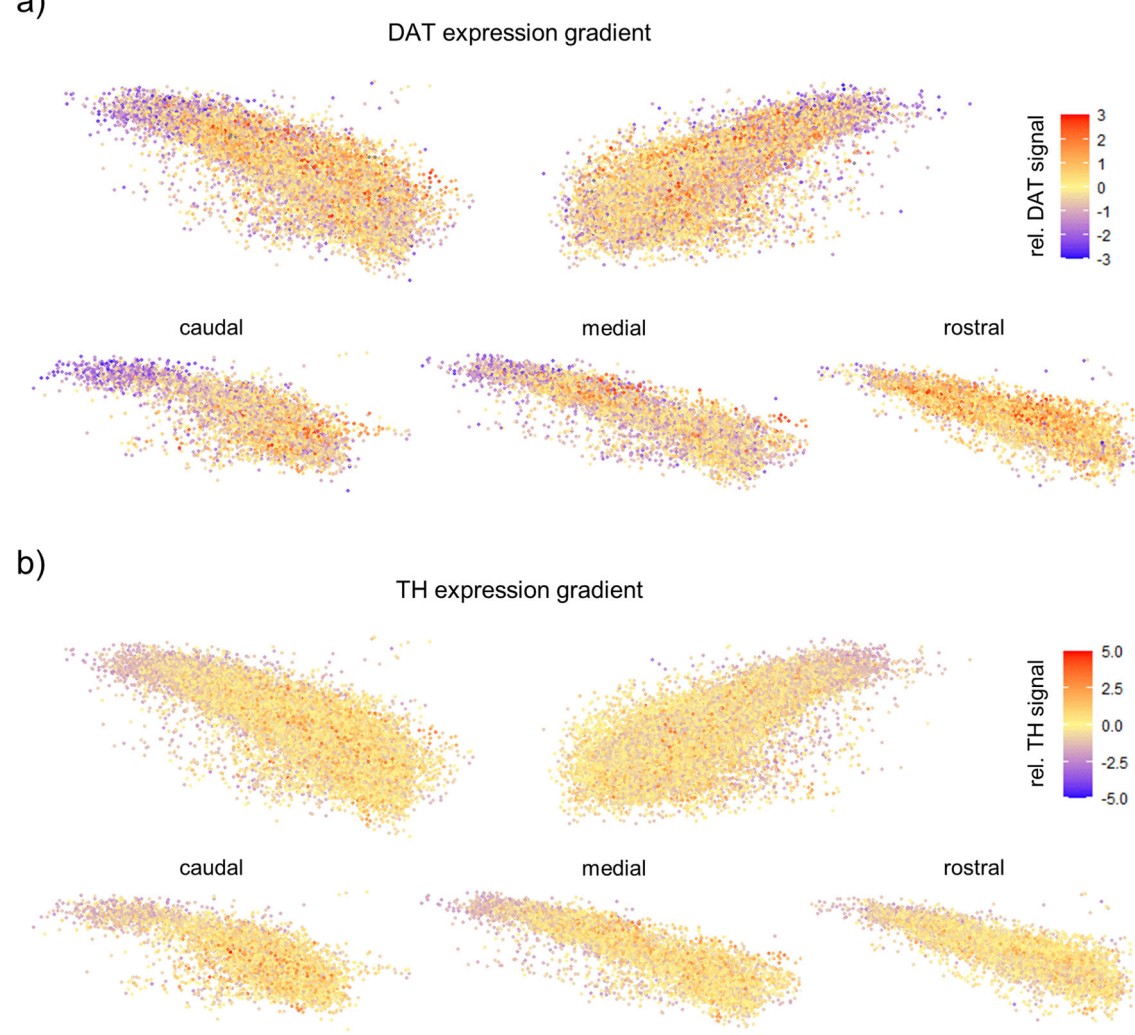

**Fig. 9 IF-derived relative TH- and DAT-protein expression in TH-positive SN neurons.** Plotted are the individual TH-positive neurons from Fig. 8, according to their scaled x,y-coordinates, determined via DLAP-5, for all analysed animals ($n = $ ~38500, $N = 14$). Scaled relative (rel.) fluorescence-values of DAT (**a**) and TH (**b**) signals are colour coded according to their individual deviation from the scaled mean-values (0.0) for each animal. Corresponding 3D maps are given in Fig. S7.

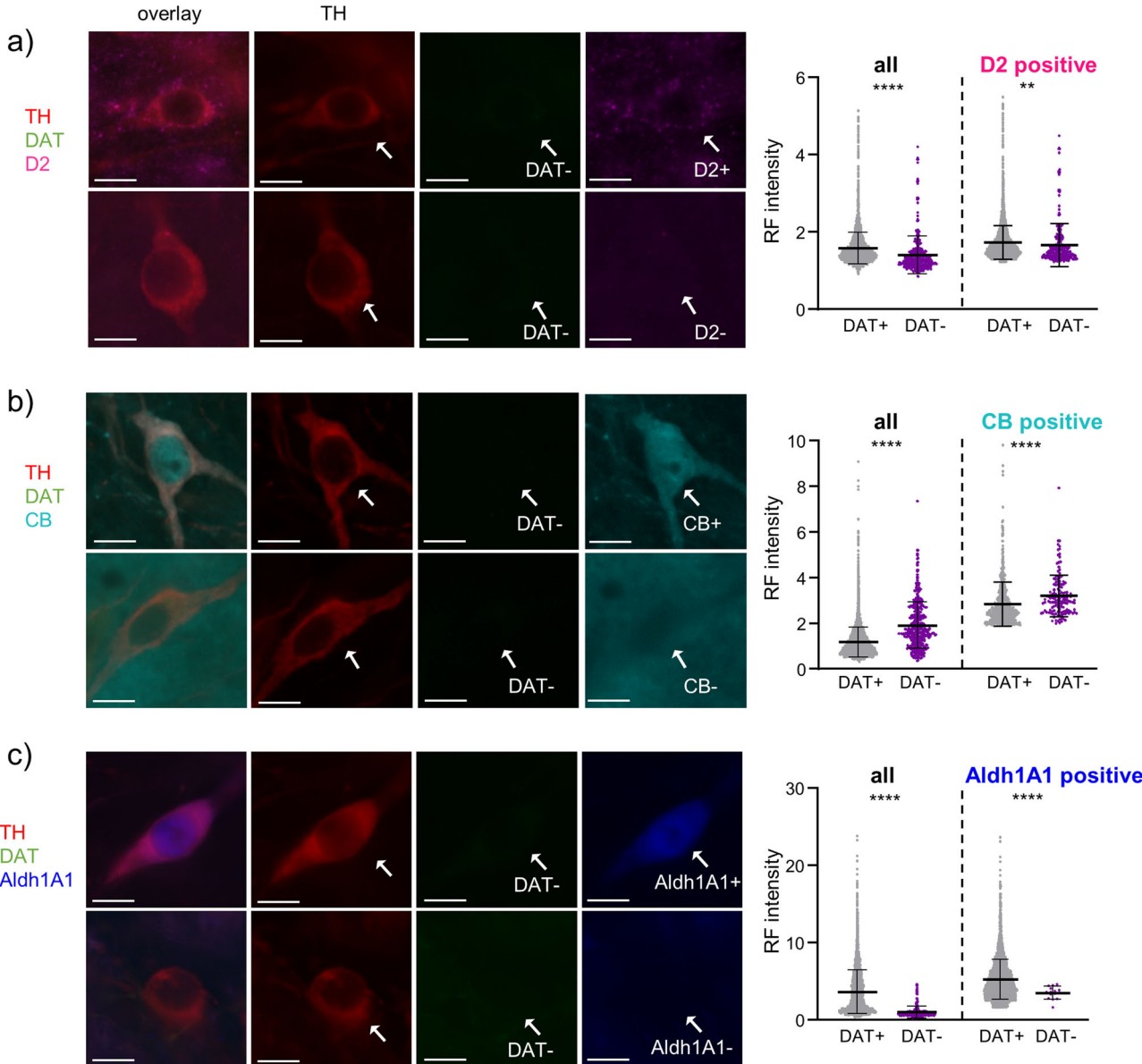

**Fig. 10 IF-derived relative expression of D2, CB, and Aldh1A1 in TH-positive DAT-negative SN neurons.** Left: SN neurons in coronal midbrain sections from adult mice after IF for tyrosine hydroxylase (TH, red), dopamine transporter (DAT, green) and (**a**) dopamine D2 autoreceptor (D2, magenta), (**b**) calbindin (CB, cyan), or (**c**) aldehyde dehydrogenase (Aldh1A1, blue). Scale bars: 10 μm. White arrows indicate TH-positive, DAT-negative neurons that are positive (top) or negative (bottom) for the respective target gene. Right: relative fluorescence (RF) signal intensities for D2, CB, and Aldh1A1 in DAT-positive (DAT+) and DAT-negative (DAT−) neurons for all analysed TH-positive neurons (left), and for those TH-positive neurons only that were also immuno-positive for D2, CB, or Aldh1A1, respectively (right). Data are given as scatterplots, and mean ± SD for all analysed animals ($N = 3$; DAT+ $n = 722$-8206, DAT− $n = 14$–383). Significant differences according to Mann–Whitney tests (**$p < 0.01$, ****$p < 0.0001$). Data are detailed in Fig. S8 and Tables S14, S15.

associated KChip3. However, in the cytoplasm, KChip3 is also known as the enzyme calsenilin, regulating presenilins, and in the nucleus, KChip3 acts as gene-transcription-repressor DREAM (downstream regulatory element antagonist modulator), illustrating the importance of subcellular localization-analysis[100,101]. A prerequisite for DLAP-6 training is the availability of a very well-suited antibody to detect and separately analyse signals in plasma-membrane, cytoplasm and nucleus of DA neurons. Therefore, we utilised a suitable Kv4.3 antibody that we and others had already used for immunoelectron-microscopy[94,95,97].

Figure 12a shows a representative confocal image of a TH-positive SN neuron in a coronal mouse brain section, after IF-staining, before automated cell body detection (input image),

and after manual as well as algorithm-based identification of the cell cytoplasm (TH, red), the nucleus (DAPI, blue), and the plasma-membrane (Kv4.3, green). As expected, the highest relative fluorescence (RF) signals for each gene were detected in these respective compartments, with manual as well as DLAP-6 analysis (RF for TH in cytoplasmic compartment: manual 55 ± 3% vs. DLAP-6 57 ± 4%, $p = 0.5862$; DAPI/nucleus: manual 76 ± 7% vs. DLAP-6 76 ± 8%, $p = 0.9270$; Kv4.3/membrane: manual 75 ± 8% vs. DLAP-6 68 ± 9%, $p < 0.0001$; Fig. 12b, Tables S18, S19).

Particularly for detecting the RF-signal in the plasma-membrane compartment, the algorithm appeared superior or more consistent/precise than the manual analysis. Accordingly, the single cell correlation manual vs DLAP-6 is less good - but

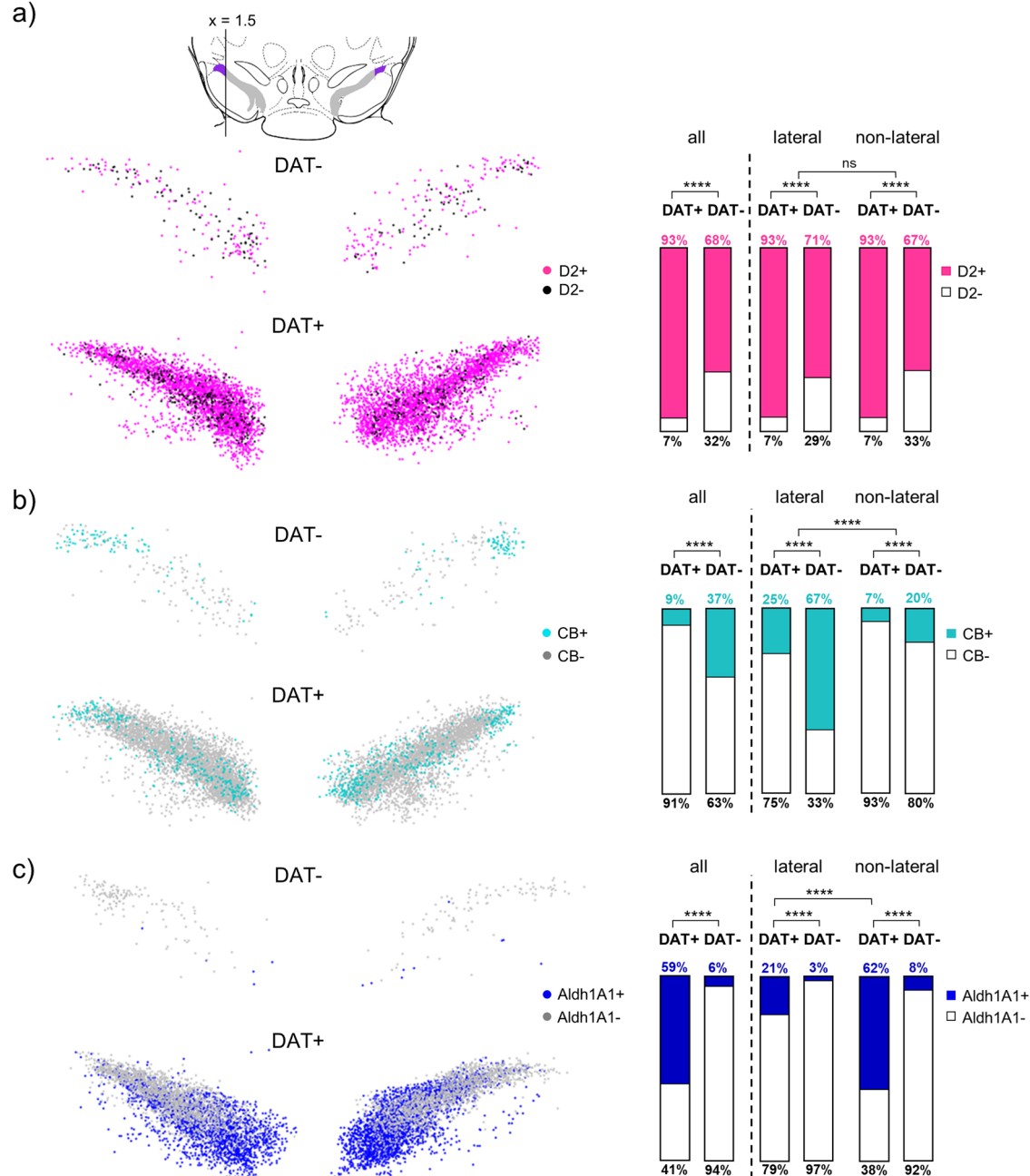

**Fig. 11 Co-expression of D2, CB, and Aldh1A1 in TH-positive SN neurons. a** Upper: illustration of the definition of the lateral SN, as in Fig. 7d, S6. Left: plotted are the respective co-expression patterns of individual TH-positive neurons, as indicated, for all analysed animals ($N = 3$; DAT+ $n = 124$–7341, DAT− $n = 3$–209), according to their scaled x,y-coordinates, determined via DLAP-5 (similar as in Fig. 7c). The resulting anatomical 2D-maps display the medio-lateral distribution of (**a**) D2, (**b**) CB, and (**c**) Aldh1A1 in the DAT-negative and DAT-positive SN neurons. Right: respective co-expression rates in all TH-positive SN neurons, and separately for the lateral and non-lateral SN, as indicated. Significant difference according to Chi square tests (****$p < 0.0001$). Data are detailed in Table S16 and Fig. S8.

still very high - ($R^2 = 0.93$ for membrane compared to 0.99 for cytoplasm and nucleus, Fig. 12c, Table S19). Most importantly, the DLAP-6 algorithm has successfully learned to identify SN DA neuron plasma-membranes, irrespective of a Kv4.3 derived fluorescence signal, and thus enables a reliable identification of the membrane-compartment in the absence of a membrane-derived fluorescence-signal.

For a direct proof, we tested the DLAP-6 approach on SN DA neurons from Kv4.3 knock-out (KO) mice (Fig. 12d, S9, Table S20). As expected, SN DA neurons from Kv4.3 KO mice did not show any Kv4.3 derived membrane-signal higher than the

background, while in wildtype (WT), mean RF-values were ~10-fold higher (DLAP-6: RF KO: $0.8 \pm 0.5$, RF WT: $7.7 \pm 2.9$, $p < 0.0001$). Nevertheless, the plasma-membrane compartment was marked by the DLAP-6 approach in TH-positive neurons from KO mice with similar robustness as for neurons from wildtype mice - a nearly impossible task by manual analysis.

## Discussion
Here, we detail a deep learning-based image analysis platform (DLAP), including six pre-designed algorithms for automated

image analysis that are easily adapted to distinct scientific needs, and allows rapid and unbiased cell count, as well as quantification of fluorescence signals (e.g., derived from mRNA or proteins) in a given ROI. These are common tasks in cell biology, but manual analysis is time consuming and prone to human error[31,32]. However, automated AI based and related approaches, like ImageJ/Fiji plugins[102] such as the Trainable Weka Segmentation[103], CellProfiler[104,105], QuPath[106], U-Net[39], and DeepImageJ[107], Cellpose[108,109], or the HALO® image analysis platform by Indica Labs[110], require significant computational skills and/or high-end hardware, a significant barrier for their routine use[111–113]. Hence, manual approaches are still widely used for image analysis in life sciences[62,114,115].

Our DLAP approach overcomes these issues by providing already pre-trained neural networks that can learn to detect new labels and extract desired features, with only a small amount of additional training-images. The six DLAP algorithms are pre-designed to suit most common life science tasks, and the flexible design allows their quick and easy adaption to other cell types, specimens, and scientific questions, by retraining them with a few additional respective images via the user-friendly web-platform. The FCN algorithm is not pre-trained, but can be easily trained from scratch, as less training data is required, because of its smaller size compared to the Deeplab3 neural network. For the loss function[116], we briefly experimented with training the FCN with the dice loss instead of the common pixel-wise cross-entropy[117], but did not find a significant improvement (see ref. [118] for a recent discussion of these loss functions in the context of medical image segmentation).

DLAP dramatically reduces the respective analysis-time, and it is straightforward to use (utilised by under-graduate students in our lab). Our approach directly combines deep learning-based segmentation with a variety of more conventional post-processing methods (such as the watershed algorithm), leading to high quality segmentation results. Moreover, as training and image-analyses are carried out via an easy-to-use web-based interface (https://wolution.ai/), and Wolution provides in-depth support for specific adjustments, our approach does not require any sophisticated hardware, software, or programming skills. Importantly, DLAP details are not proprietary, but all steps and strategies are provided here for free and in full detail, to facilitate distribution and use within the scientific community. These features represent important advantages in comparison to other cloud-based analysis pipelines, for example, Aiforia[54,74], CDeep3M[119], or Visiopharm[120] that do not disclose insights into the underlying algorithms, do not allow user-based modifications, or require advanced programming experience (e.g., DeepCell Kiosk or Cellpose[109,121]). For example, the recent freely distributed well-suited algorithm for cell-segmentation, Cellpose2.0 did enable equally proper segmentation of TH-positive neurons from our IF-images, as our DLAP algorithms. However, in contrast to DLAP, it does not enable any automated post-processing or post-analysis of segmented ROIs for further downstream analysis (like whatershed algorithms, threshold based background exclusion, or quantification of fluorescent dot numbers or intensities) in one single package, but the user would require to write additional scripts for such downstream-application. Hence, as Cellpose2.0 relies on Python packages, despite its Graphical User Interface, basic programming knowledge is required, e.g., for troubleshooting package related issues or run-time errors. One particular additional advantage of our DLAP-6 is that it automatically segments the cellular membrane compartment, in the absence of any membrane marker, and thus enables a more detailed sub-cellular analysis.

The here detailed DLAP approach allows the systematic quantitative analysis and anatomical mapping of several

thousands of neurons in reasonable time, and thus facilitates the identification and characterization of small cellular subpopulations. Accordingly, by DLAP-analysis of about 40,000 TH-immuno-positive SN neurons in PFA-fixed brain-sections from adult mice, and by generation of expression-maps according to anatomical coordinates, we defined a small subpopulation of neurons (~5% of all TH-positive SN neurons). These neurons were immuno-negative for the dopamine transporter DAT, were mainly located in the caudo-lateral SN, and had ~40% smaller cell bodies. Due to their localization in the lateral SN, we can exclude that the DAT-negative TH-positive neurons are VTA neurons[92]. However, we do not exclude that they are non-dopaminergic, as TH-expression alone is not a proof of neurons being dopaminergic[2]. TH-positive but DAT negative striatal inter-neurons have been described[122]. However, interneurons in the SN are rare[123,124] and rather small (~three times smaller than classical DAT-positive CB-negative SN DA neurons,[125]. Given the still relatively large cell body size of the DAT-negative lateral SN neurons (~120 μm$^2$ compared to ~195 cm$^2$), here we would not unequivocally conclude that these are interneurons, while we do not rule out this possibility.

DAT is a major determinant of activity and excitability of DA neurons, and of dopamine homeostasis and transmission[44,45,126]. It is inhibited e.g., by methylphenidate and amphetamines as treatment for ADHD and depression[19,127], and changes in DAT expression have been reported in schizophrenia, ADHD, and Parkinson's[128]. However beyond disease states, up to now, only for VTA DA neurons lower DAT expression has been analysed[78,79,82]. On the contrary, for SN DA neurons, DAT is commonly used as specific marker besides TH[93,129], as well as for their targeting, for instance, via DAT-Cre transgenic mouse lines, expressing the Cre-recombinase under the DAT-promotor[47–49,92,130]. Our results imply that with such approaches, target-gene expression might not be affected in those lateral TH-positive SN neurons that are immuno-negative for DAT protein. However, it must be noted that we did not perform an absolute DAT-protein quantification. Hence, we explicitly do not exclude very low expression of DAT protein in the plasma-membranes of these lateral SN neurons, within the range of the respective detected DAT-background immuno-signals (similar applies for D2, CB and Aldh1A1). Such a low DAT expression would still be sufficient for successful DAT-Cre recombination[131,132].

We found that the DAT-negative TH-positive SN neurons in the lateral SN were negative for Aldh1A1, a marker for more vulnerable, (medio-)ventral tier SN DA neurons, while the co-expression rate was massively (~7-fold; ~70% vs 8%) increased for CB, a marker for less vulnerable SN DA neurons. For classical DAT- and D2-AR-positive SN DA neurons, CB expression and/or absence of Aldh1A1 is a marker for less vulnerable neurons, suggesting that the non-classic DAT-negative neurons in the caudo-lateral SN might be less vulnerable as well. However, the absence of inhibitory D2-AR in ~30% of DAT-negative SN neurons might render them more vulnerable toward excitotoxicity, as their activity control by dopamine is missing[57,133]. On the other hand, the mesocortical VTA DA neurons, that are hardly affected in PD, express no functional D2-AR[82,134]. For more lateral SN DA neurons, a higher sensitivity towards PD-stressors is described[15,135–138]. In terms of axonal projections, it is described that the (caudo-) lateral SN DA neurons project into the (rostro-) dorsolateral striatum[139], and it corresponds to the ventro-lateral SN DA neurons in humans[140,141]. These neurons have been described to display a higher vulnerability in Parkinson's compared to rostro-medial SN DA neurons[15,142–146]. However, whether these neurons are DAT-positive or DAT-negative has not yet been analysed, and in general, there is no clear correlation between DAT-expression and vulnerability[147],

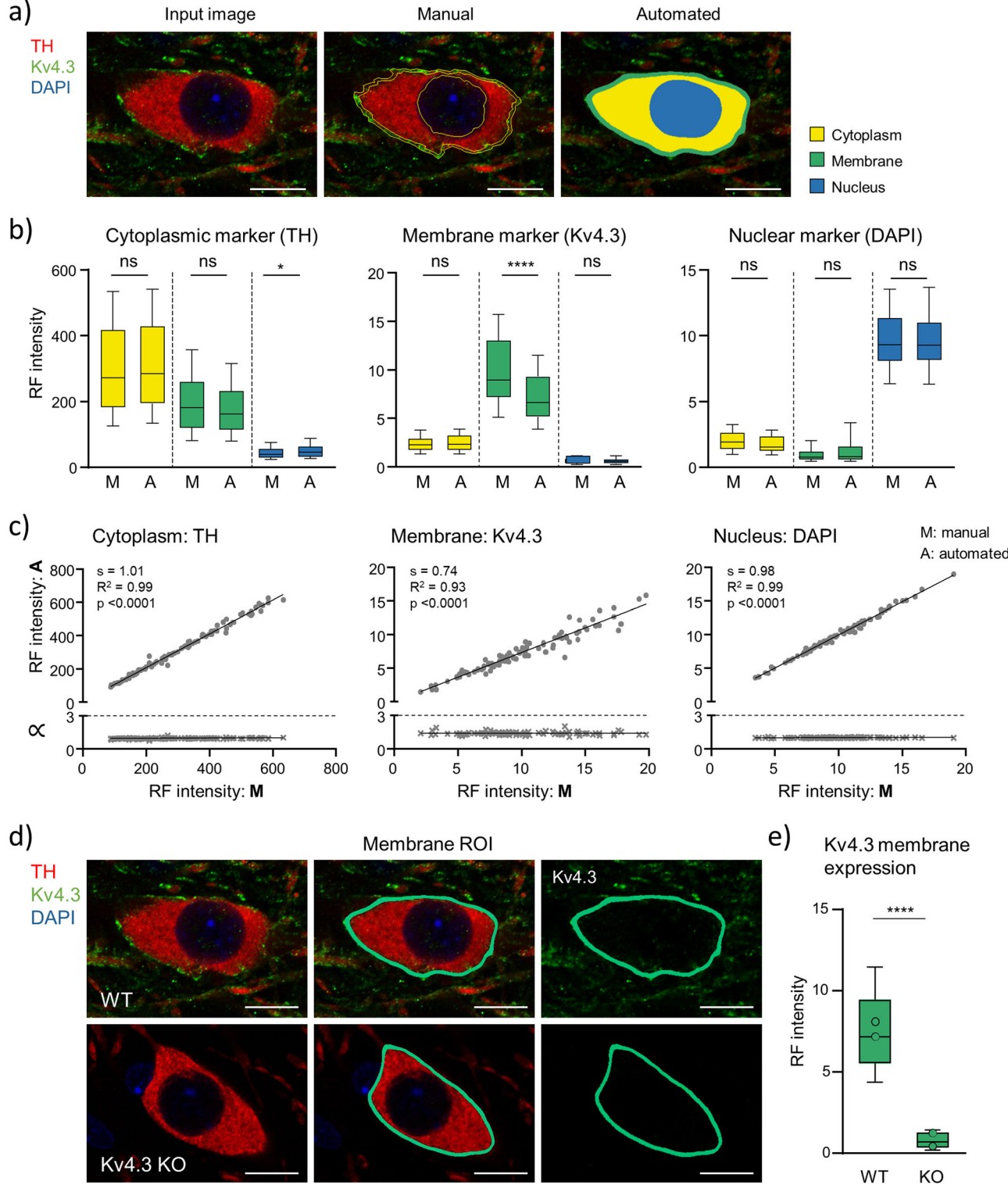

**Fig. 12 Comparison between manual and automated (DLAP-6) IF-derived image analysis in cellular compartments of TH-positive SN neurons. a** SN DA neurons in coronal midbrain section from adult mice after IF for tyrosine hydroxylase (TH, red) and the $K^+$ channel α-subunit Kv4.3 (green). Nuclei are marked by DAPI-staining (blue). Left: input image. Middle: image after manual labelling of cell body, nucleus and membrane. Right: image after automated recognition of the TH-labelled cell body (yellow area), nucleus (blue area) and membrane (green area) with DLAP-6. Scale bars: 10 μm. **b** Relative fluorescence (RF) intensity/cell in different sub-cellular compartments, quantified manually (M) or automatically (A), as indicated. Data are given as boxplots (median, 10–90 percentile) for all analysed neurons ($n = 94$). Significant differences according to Mann–Whitney tests (ns: $p > 0.05$, ****$p < 0.0001$). **c** Upper: single neuron correlations of RF intensities/cell between manual and automated analysis for all three marker genes in their respective compartment according to Pearson correlation test. Lower: corresponding proportionality constants ∝, calculated from the manual and automated signal intensity ratios. **d** SN DA neurons in coronal midbrain section from adult WT (top) and Kv4.3 KO (bottom) mice after IF for TH and Kv4.3, similar as in (**a**). **e** Kv4.3 RF intensity/neuron-membrane compartment in WT and KO mice, as indicated. Data are given as boxplots (median, 10–90 percentile) for all analysed neuros (WT: $n = 179$, Kv4.3 KO: $n = 200$). Open circles indicate mean values for each analysed animal. Significant difference according to Mann–Whitney test (****$p < 0.0001$). All data are detailed in Tables S18–20 and Fig. S9.

suggested in earlier studies[148,149]. Hence, additional studies are necessary to address the vulnerability of the here identified DAT-negative caudo-lateral SN neurons.

Very lateral SN DA neurons have been reported to be positive for the vesicular glutamate transporter Vglut2, and rather negative for the transcription factor SOX6[87,92,93]. In line, our preliminary single nuclei sequencing data indicate that the here described DAT-negative SN neurons are also Vglut2-positive, while SOX6 expression is at least significantly reduced, and they confirmed the absence of Aldh1A1 in TH-positive and DAT-negative SN neurons at the mRNA level. In accordance with this and the high degree of CB co-expression, in a similar approach, it recently has been shown that one type of SN DA neuron clusters preferentially expressed SOX6 mRNA while other clusters preferentially expressed CB[150].

What is known about the function of lateral SN DA neurons? Vglut2- and CB-positive but Aldh1A1-negative lateral SN DA neurons are important for responses to novel cues and salience[87,91,129,130,151]. Sox6-negative SN DA neurons have been shown to project to the medial, ventral, and caudal striatum and respond to rewards[152]. Some lateral SN DA neurons are activated by aversive stimuli or cues that predict aversive stimuli[153]. However, these studies either analysed only DAT-positive neurons or they did not report whether the analysed neurons were DAT-positive or -negative. Lateral SN DA neurons in vivo display a higher burst activity (that is particularly metabolically demanding) compared to medial SN DA neurons[154]. In vitro, the pacemaker frequency of lateral but not medial SN DA neurons is positively coupled to the activity of Cav1.3 L-type voltage gated $Ca^{2+}$ channels[155]. These channels have been linked to the vulnerabilities of SN DA neurons in PD[57,58,156,157]. However, it should be noted that the definition of the "lateral SN" in this - and most of the studies cited above - is much broader, and likely includes the here defined very lateral TH-positive SN neurons (compare Fig. S6), but also more medial, DAT-positive neurons in the lateral SN. One recent study has indeed addressed the electrophysiological properties exclusively of only very lateral SN DA neurons, expressing CB or Vglut2, most likely similar to the very lateral DAT immuno-negative neuronal population that we describe here. The cell bodies of these neurons were also significantly smaller than those of classical SN DA neurons, had a higher input resistance and excitability, and they displayed a less precise pacemaker, with lower frequencies and less prominent after-hyperpolarisations, compared to classical SN DA neurons[158]. However, future studies are necessary to fully characterize the here defined and quantified subpopulation of lateral TH-positive SN DA, in terms of their distinct molecular identity, their axonal projections, their physiological function, and their fate in disease.

## Methods

**DLAP algorithms and underlying convolutional neural networks**. We have used two very different neuronal network architectures, based on either the DeepLab3 network architecture or on a fully convolutional neural network (FCN). DeepLab3 based networks are more powerful for detecting complicated larger objects, like complex neuronal structures, and beyond, while FCN perform in general better for fine structures (like RNAscope probe derived, small individual dots), rather than large global concepts[41,159]. Both types of neural networks belong to the class of convolutional neuronal networks (CNN), commonly used for image analysis[37]. Convolutional layers do not receive their inputs from the whole previous layer but only from a certain area (called 'filter' or 'kernel') which increases their computational efficiency[160]. Moreover, they assign to each pixel in the image a

class, such as cellular nucleus or background. This type of analysis is called semantic segmentation. The performance of deep learning based semantic segmentation depends strongly on the neural network architecture, training parameters and in particular the quality of the training dataset. By specific training and further optimizing these two network-types for distinct tasks, we generated six distinct algorithms, as specified in Tables 1, 2.

For cell counting and immunofluorescence signal quantification (DLAP-3 to 6), the large pre-trained neural network DeepLab3[40] was further trained and optimized post training for the respective specific tasks. DeepLab3 is one of the best performing image segmentation algorithms on the Pascal Visual Object Classes (VOC) dataset[161]. The Pascal VOC is a standardized image dataset for building and evaluating algorithms, often used as a benchmark for segmentation quality. In general, the key innovation of DeepLab algorithms is the use of so called dilated or atrous convolutions. This type of convolutions has a dilated filter size, increasing the area covered by each filter, and the context that can be incorporated. Therefore, they have a higher resolution and can integrate information over larger areas, while keeping the number of parameters of the convolutions the same[159]. Specifically, for our algorithms 3 to 6, we used xception65 as the CNN backend, provided by Google [https://github.com/tensorflow/models/tree/master/research/deeplab][162], which was pre-trained on the Pascal VOC 2012 segmentation dataset (Everingham et al.[51]).

For analysis of RNAscope data, we developed a more simple, better-suited FCN (DLAP-1 and 2), as the trained respective algorithms based on DeepLab3 were not optimal for reliable detection of RNAscope-derived dot-numbers (Table S1). The most important modification of the FCN architecture is that it uses only convolutions, rather than including fully connected linear layers[41]. Our FCN has a smaller receptive field and conserves image resolution at each convolutional layer. Replacing the linear layers with fixed dimensions by convolutions allows the network to accept arbitrarily large images for processing and making predictions (called interference), and to process them very efficiently - limited mainly by the graphics processing unit (GPU) memory size. The FCN architecture, illustrated in Fig. 2, is our own development, using the Tensorflow library [https://www.tensorflow.org/]. Our FCN network uses five convolutional layers, connected by the Rectified Linear Units (ReLU) activation function. This function only transfers the direct output of the previous layer to the next layer, if it is positive, while it returns zero for negative outputs[163]. The softmax activation function normalises the network output to the probability distribution. Our network does not include any downsampling/downsizing to reduce image resolution through dropout or strided (kernel step size > 1) convolutions. Therefore, the output image has the same resolution as the input image, and fine structures are not smoothed out. We used another approach of neural network architectures, the so-called 1x1 convolutions, to reduce the number of feature maps and keep the parameter number tractable[164]. Such a convolution is used between feature map 2 and feature map 3 to reduce the number of kernels by a factor of 4, thus reducing the parameter size of the following convolution by a factor of 16. To conserve image dimension, we extend the outer frame at the boundary of each image, which increases the space for the filter/kernel to scan the image, a process also called mirror padding. This architecture provided a good compromise between simplicity (and thus computational speed) and segmentation quality.

**Algorithm training and post processing parameters**. Training datasets need to have sufficient quality and size, depending on

data variability[112,165]. However, we observed that using too large training datasets leads to overfitting, which resulted in decreased generalization. We ensured the quality (e.g., resolution, frame size) of our training datasets, and performed a quality evaluation during training (see below), in addition to data-validation by comparison with manual analyses. Images showing different neuronal shapes/forms in different brightness intensities were used for training to ensure the identification and recognition over a vast array of brightness and cellular structure.

As our FCN is not based on a pre-trained neural network (no "transfer learning"), it was trained by us from scratch, by only using our own training data. This is possible because of its smaller size compared to the Deeplab neural network, which also means that less training data is required. The FCN-based algorithms 1 and 2 analyse image patches of size $63 \times 63$ pixels. To train the network, we randomly cropped the images from our manually labelled training dataset (ground truth data) into such patches. Training was performed using the stochastic first-order gradient-based Adam optimizer with default settings (learning rate $\alpha = 0.001$, estimates decay rates $\beta_1 = 0.9$, $\beta_2 = 0.999$, to prevent division by zero $\varepsilon = 10^{-8}$)[166]. For DeepLab3 based algorithms DLAP-3 to 6, we adjusted the neural network weights by further training with our own datasets using the training script provided by Google Research[162]. This was performed on randomly sampled patches of size $512^2$ pixels, much larger than the $63^2$ pixels we use in the FCN algorithm, as Deeplab3 is designed to detect large, complex objects within an image.

For both network-types, the training time on a modern GPU was several hours for each of the datasets. The number of images and neurons each that were used to train each of the individual algorithms is given in Table 2. We used 80% of manually labelled images for the training dataset, and the remaining 20% of the images for the "test data" set. Train-test-split on the level of images was chosen to avoid data leakage. For this performance-evaluation of the individual algorithms, the "test data" was analysed and the results were compared to the ground truth provided by the labels. For all algorithms, we assessed typical quality measures (Table 1, S1): The pixel error rate, i.e., the global percentage of pixels that are wrongly classified. This global error rate was calculated by defining for each segmentation class the rates of true positives (TP), true negatives (TN), false positives (FP) and false negatives (FN), each given in % of the total pixels of the whole training dataset. The specificity [defined as TN/(TN + FP)] and the sensitivity [defined as TP/(TP + FN)], for each segmentation class, were also calculated, as these values are more suitable measures in cases of large class imbalance.

We trained each dataset until the training error/loss function (difference between output and ground truth) converged to a minimum. We also use "Early Stopping" in the training process[167] to avoid overtraining (see above). We verified that we did not overtrain, by tracking the pixel error in the "test data". If overtraining is happening, only the training error is still getting lower, while the test error increases. Training was stopped as soon as the pixel error was getting worse not better. Performances of all algorithms were evaluated on the respective "test data" (compare Table 2).

For post training image analysis optimization, the quality of the CNN outputs for each task (summarized in Table 1) was further improved by, implementing additional classic image analysis procedures (a) to (e), specified below, into the individual CNN procedures. If an analysis step was further increasing the correlation between algorithm- and manually-obtained results for a distinct task, it was implemented into the respective algorithm. The specific steps (a-e) that were implemented post-training into each of the final algorithms are summarized in Table 2.

a.  Watershed algorithm requires a set of markers, which are associated with the centres of respective target structures[168]. These markers are obtained by calculating the Euclidean distance transform of the binary mask of target structures (cells detected by the neural network), and then finding peaks in the distance image. Starting from these markers, the watershed algorithm "floods" the image until all pixels are assigned with a watershed region. We used this algorithm in both FCN based algorithms, for a better separation of mRNA molecule derived dots. The FCN output tend to connect single fluorescent dots of individual mRNA molecules, which biases their number count. We found that, by applying a classical watershed algorithm we can separate these points and get a number count much closer to the truth (manual count). For the DeepLab3 based algorithms 4 and 5, watershed algorithm were used to improve segmentation of neighbouring nuclei/cells. This was not necessary for DLAP-3.

b.  Morphological operations like type dilation and erosion[169] were applied to algorithms 1 and 2. These operations add or remove pixels from the object boundaries. This improves separation of individual neurons and smoothens out uneven boundaries provided by the CNN. This approach was used in both FCN-based algorithms for a better separation of individual neurons.

c.  The Binary fill-holes-function was applied to the DeepLab3-based CNN outputs for DLAP-4 to 6, where pixel-wise semantic segmentation resulted in holes within the detected compartments, as such holes are of no biological meaning. Therefore, all holes in the segmentation masks of each region were filled using the "binary_fill_-holes" function[147] of the SciPy library http://scipy.github.io/devdocs/reference/generated/scipy.ndimage.binary_fill_holes.html.

d.  A procedure to remove target gene colour channel was applied to the DeepLab3 based algorithms 5 and 6 to facilitate identification of the target areas within the ROI (i.e., neuronal cell bodies or neuronal compartments), in which the relative target gene derived fluorescence-signal was quantified. This procedure is removing the target gene channel for the algorithm-based detection of the target areas, to ensure its recognition, independently of the target gene fluorescence-signal. For excluding the target gene channel, the respective colour layer of the image was set to zero before passing the image through the CNN (both for training and analysis), and it was added back, for the target-signal quantification step of the algorithms.

e.  An RGB intensity threshold was included to the DeepLab3 based algorithms 5 and 6, to optimize background intensity quantification for calculation of relative target gene signal intensities in identified target areas (i.e., neuronal cell bodies or neuronal compartments) to correct for non-uniform intensity conditions in different images/experiments. It was not always straightforward to determine the individual background intensities, because some images contained large regions with no detected target areas, but also with no apparent blank background. We included RGB intensity thresholds (for widefield and for confocal images) of pixel values ranging from 10-40 for each channel, depending on its raw intensities, to exclude pixels higher than this threshold from determining background signal intensities.

**Mice**. Juvenile (~PN13, Fig. 5 data), and adult (~PN90) male mice were bred at Ulm University at a 12-h light/12-h dark cycle

and were fed ad libitum. Kv4.3 KO mice were obtained from Jean-Marc Goaillard[97]. All animal procedures were approved by the German Regierungspräsidium Tübingen (Ref: 35/9185.81-3; TV-No. 1043, Reg. Nr. o.147) and carried out in accordance with the approved guidelines.

*Drug and saline treated animals* (Fig. 4 data), were the same, cohort as already published (Cav2.3 +/+ mice from Benkert, Hess[54]). Drug treated mice were treated in vivo with MPTP/ probenecid to introduce SN DA neuron degeneration. Briefly, MPTP hydrochloride was injected 10 times (every 3.5 days for 5 weeks) subcutaneously at a concentration of 20 mg/kg saline (sigma) together with probenecid intraperitoneally at a concentration of 250 mg/kg in 1x PBS (Thermo Fisher). Control mice were injected with saline and probenecid. For all details, including, TH-DAB/hematoxillin staining, stereology, and Aforia-based automated analysis, (see Benkert, Hess[54]).

**Human brain samples**. Human midbrain tissue including the Substantia nigra (SN) was collected and obtained from the German Brain Bank (www.brain-net.net), Grant-No. GA76 and GA82), as native cryo-preserved tissue blocks (−80 °C/dry ice). Analysis of the human material was approved by the ethic commission of the German Brain Bank as well as of Ulm University (277/07-UBB/se). Informed consent from all participants was given to the German Brain Bank. Detailed information on human midbrain samples, including sex, is summarized in Table S5. We have no information if/how the information regarding sex and gender was obtained. Human tissue was processed, stained, and the RNA integrity was assessed via determining the RNA integrity number (RIN), using the Agilent 2100 Bioanalyzer as described[59].

**RNAscope® in situ hybridization, image acquisition, and data analysis**. *RNAscope experiments* were performed essentially as described[54,170]. *The* RNAscope® technology (Advanced Cell Diagnostics, ACD) was performed according to the provided protocol for fresh frozen tissue sections [https://acdbio.com]. Details of the used RNAscope probes are given in Table S2. All used chemicals were RNAse free grade.

12 μm coronal cryosections of mouse or human midbrains were prepared, using a Leica CM3050S cryotome as previously described[59,171], mounted on SuperFrost® Plus glass slides (VWR), and allowed to dry in a drying chamber containing silica gel (Merck) at −20 °C for 1 h. After fixation with 4% PFA (Thermo Fisher Scientific) in 1x PBS (pH 7.4) for 15 min at 4 °C and dehydration via an increasing ethanol series (50%, 75%, 100%, 100%, Sigma), sections were permeabilized for 30 min by digestion with protease IV (ACD) at room temperature (RT). Following digestion, respective RNAscope probes (TH probes and target gene probes were processed in parallel) were hybridized for 2 h at 40 °C in a HybEZ II hybridization oven (ACD). Signal was amplified using the RNAscope Fluorescent Multiplex Detection Kit (ACD) containing four amplification probes (AMP1-4). In between each amplification step (incubation at 40 °C for 30 min with AMP1/3 or 15 min with AMP2/4), sections were washed twice for 2 min with wash buffer (ACD). After RNAscope hybridization, sections were counterstained with 4′,6-diamidino-2-phenylindole (DAPI) ready-to-use solution (ACD, included in Kit) for 30 s at RT. Sides were coverslipped with HardSet mounting medium (VectaShield), and allowed to dry overnight at 4 °C. RNAscope probes for TH were labelled with AlexaFluor488, and target gene probes with Atto550.

Fluorescent images containing the ROI (i.e., the SN), of murine and human midbrain sections were acquired at 63x magnification using a Leica DM6 B epifluorescence microscope. All images were

acquired as Z-stacks, covering the full depth of cells, and reduced to maximum intensity Z-projections, by using Fiji [http://imagej. net/Fiji]. TH-derived RNAscope fluorescence signals were visualized at 480/40 nm excitation, 527/30 nm emission and 505 nm dichroic mirror, target gene-derived fluorescence signals were visualized at 546/10 nm excitation, 585/40 nm emission and 560 nm dichroic mirror, DAPI-signals were visualized at 350/ 50 nm excitation, 460/50 nm emission and 400 nm dichroic mirror. Files were aquired using the LASX software (Leica), and stored as PNG files.

For manual image analysis, the images were further processed using the Fiji software [http://imagej.net/Fiji]. First, cell bodies of individual TH- and DAPI-positive neurons were encircled, using the "Polygon selection" tool to define the area for RNAscope signal quantification. In the next step, a classifier was trained on images only showing target gene signal, by manually labelling and annotating background and dot regions (around 30 labels for each class) in the "Trainable Weka Segmentation". This classifier was then used to classify all target gene images into the two groups "dots" and "background". Afterwards, the threshold for particle recognition was set on classified images, and dots were counted for each cell-specific ROI using the "analyse particle" function. Target probe hybridization results in a small fluorescent dot for each mRNA molecule, allowing absolute quantification of mRNA molecules (via dot counting) independent from fluorescent signal intensity.

For automated quantification of RNAscope signals, custom-designed FCN based algorithms DLAP-1 (for mouse-brain sections) and 2 (for human brain sections) and the Wolution-platform were used [https://console.wolution.ai/] (Wolution GmbH & Co. KG, Planegg, Germany). Processed images were uploaded on the Wolution platform, and the algorithms automatically marked cell bodies of TH-positive cells and quantified the number (and the dot area) of target gene derived fluorescence dots. After algorithm processing, correct identification of cells was controlled by hand and resulted in inclusion of ~50% correctly identified cells for analysis. If a quantification of all TH-positive neurons is required, neuronal nuclei must be counted, before excluding neurons with not optimally separated cell bodies for subsequent RNAscope based mRNA quantification in the target cells.

We carried out a statistical modelling for brightfield light adjustment to asses and correct for the possible influence of neuromelanin (NM) in human-derived RNAscope data applying a linear mixed effect model, similar as we had previously described[59,60]. Briefly, we applied a bayesian probabilistic model approach conducted in RStan[172] with the R package version 2.21.7, https://mc-stan.org/) using R (version 4.2) under RStudio (version 2022.07). We applied normal distributions for priors of all parameters. To achieve a stable non-degenerate solution priors needed to be properly informed by restricting the effective parameter space. The chosen model, detailed below, is well-informed by the data on all its parameters and shows an acceptable fit on the age level (see posterior-predictive plots in Fig. S2).

The number of detected dots/mRNA-molecules for each component/gene (defined as $Y_G$) is assumed to follow a binomial distribution in which the trial size is derived from cell areas:

$$Y_G \sim Binom(s, p_G) \qquad (1)$$

The probability pG of detecting a target molecule is expressed according to a logit-link function (linear combination) as:

$$p_G = \frac{p_{\max}}{1 + e^{-g(x)}} \qquad (2)$$

where $g(x)$ is an equation, indexed by the mixture-component and a binary discretization of a brain donor (adult vs. aged) that

considers a cell's NM concentration (as given by brightfield light transmission, $T$) according to the Lambert-Beer law which states that concentration is proportional to the negative logarithm of light transmission

$$-\log_{10} T = l\varepsilon \cdot c \qquad (3)$$

Light transmission $T$ is the fraction of transmitted to incident light intensity

$$T = \frac{I_t}{I_i} \qquad (4)$$

Since $I_t < I_i$ and light intensities are strictly positive, it follows that $0 < T < 1$. For the probabilistic model, a distribution for T has to be defined. A continuous distribution bounded on [0, 1] is the Beta-distribution.

$$T \sim Beta(\mu_T, \kappa) \qquad (5)$$

The Beta distribution is parameterized by its mean $\mu_T$ and the precision $\kappa$.

For $\mu_T$ we apply a logit-link function:

$$\mu_T = \frac{1}{1 + e^{-h(x)}} \qquad (6)$$

$h(x)$ is a linear equation, indexed by the mixture component and binarized age. In addition, the intercept considers a random effect on individual brains. The mixture proportion is modelled to depend on binarized age and to consider a random effect on the brain level, given in Wilcoxon notation[173]

$$\theta = logit^{-1}(\theta_0 + \Delta_{\theta,age} + (1|brain)) \qquad (7)$$

To adjust experimental results for a mutual NM influence, data were fitted and then reproduced from the model, with each cell's $T$ set to a constant value according to the mean of all brains' $\mu_T$. The influence of BF light transmission (associated to NM content by $-\log T$) on gene expression / dot detection probability can be considered significant with 95%-credible intervals from $-5.1$ to $-1.5$ or from $-12.2$ to $-7.7$ for low and high expressing cells, respectively (corresponding intervals from the prior distribution vary from $-7.8$ to $+8.0$).

**Immunohistochemistry, image acquisition, and data analysis**. DAB Immunohistochemistry (IHC) experimental procedures were performed essentially as previously described[54]. Briefly, mice were transcardially perfused with 1x PBS-heparin (37 °C) for 2 min followed by ice cold 4% PFA for 4 min with a flow rate of 6 ml/min and post-fixed (overnight at 4 °C) in 4% PFA (Thermo Fisher Scientific) in 1x PBS (pH 7.4). Brains were stored in 0.05% NaN$_3$ (Sigma) in PBS at 4 °C until vibratome (VT 1000 S, Leica) cutting (30 μm coronal midbrain sections). All washing and incubation steps during the staining procedure were performed while shaking (300 rpm, microplate shaker, VWR). Free-floating sections were washed (three times in 1x PBS for 10 min) and blocked for 2 h with 10% normal goat serum (NGS, Vector Laboratories), 0.2% BSA (Carl Roth) and 0.5% Triton X-100 (Merck) in 1x PBS to prevent non-specific antibody binding. After further washing (one time in 1x PBS for 10 min), sections were incubated with rabbit anti-TH primary antibody (1:5000, Merck) in 1% NGS, 0.2% BSA and 0.5% Triton X-100 (in 1x PBS) overnight at room temperature (RT). Sections were then washed three times in 0.2% Triton X-100 in 1x PBS for 10 min and incubated with biotinylated goat anti-rabbit (1:1000, Vector Laboratories) for 2 h at room temperature. Immunostaining was visualized via VECTASTAIN® ABC system based on Horseradish peroxidase (HRP) detection (Vector Laboratories) using 3,3'-Diaminobenzidine (DAB, Vector Laboratories) as substrate. The slices were mounted on SuperFrost® Plus glass slides (VWR),

dehydrated in ascending ethanol series (50%, 70%, 90%, 100%, 100%, Sigma) for 10 min each, and cleared with xylene (Sigma) two times for 10 min each. Slides were mounted with Vecta-Mount Permanent Mounting Medium (Vector Laboratories).

For hematoxylin-counterstaining of already DAB-stained and permanently mounted adult mouse brain sections, slides were incubated in xylene (two times for 5 min) to remove coverslips and rehydrated by an ethanol series (100%, 100%, 90%, 70%, 50%) and H$_2$O (5 min each). After drying for 5 min, slides were incubated with Vector hematoxylin QS (2 min) to counterstain nuclei (Vector Hematoxylin QS Kit, Vector Laboratories). After dehydration in ethanol series and clearing in xylene they were again mounted using VectaMount (all steps as described for DAB-staining).

Stereological estimates of TH-positive neuron numbers were determined using a Leica CTR5500 microscope and the unbiased optical fractionator method (StereoInvestigator software; MBF Bioscience), similar as previously described[54,73]. The SN region throughout the whole caudo-rostral SN axis (Bregma −3.8 to −2.7, according to[50] was identified, using well established landmarks[78]), and was marked as ROI for TH-positive neuron count. The ROI was marked and analysed unilaterally on each of the serial DAB-stained sections (37 for juvenile, 40 for adult mice). Sampling grid dimensions were $75 \times 75$ μm (x,y-axes), counting frame size was $50 \times 50$ μm (x,y-axes), and counting frame height was 9 or 11 μm for juvenile and adult mice, respectively. Estimated total number of TH-positive neurons (N) was calculated for each animal according to Eq. (8):

$$N = \sum Q^- \cdot \frac{t}{h \cdot asf \cdot ssf} \qquad (8)$$

with $\sum Q^-$ = number of counted neurons, t = mean mounted section thickness (i.e., ~10–11 μm), h = counting frame height (i.e., 80% of the mounted section thickness), asf = area sampling fraction (i.e., 0.44), and ssf = section sampling fraction (i.e., 1 for SN).

Reliability of the estimation was evaluated by the Gundersen coefficient (CE, m = 1) according to Eq. (9). CE values were all ≤0.05 for all analysed animals.

$$CE = \frac{\sqrt{s^2 + VAR_{SRS}}}{s^2} \qquad (9)$$

with $A = \sum_{i=1}^{n}(Q_i^-)^2$; $B = \sum_{i=1}^{n-1} Q_i^- Q_{i+1}^-$; $C = \sum_{i=1}^{n-2} Q_i^- Q_{i+2}^-$
$s^2$ = variance due to noise, and $VAR_{SRS}$ = variance due to systematic random sampling, according to Eq. (10) for m = 1.

$$VAR_{SRS} = \frac{3(A - s^2) - 4B + C}{240} \qquad (10)$$

For automated neuron counting digital images of DAB-stained sections were acquired using a whole slide scanner (3D-Histech Pannoramic 250 Flash III, Sysmex Deutschland GmbH, Norderstedt, Germany or Aperio Versa 8, Leica Biosystems Nussloch GmbH, Nußloch, Germany) and processed using the Fiji software [http://imagej.net/Fiji]. The digital slides were processed using QuPath [https://qupath.github.io/] to label and cut out the sections of interest (37/40 consecutive SN sections covering the entire caudo-rostral axis). As for stereology, the SN ROI was identified according to the typical landmarks[50,78] and analysed unilaterally. Processed sections (showing the ROIs) were afterwards uploaded on the Aiforia® Cloud platform [https://www.aiforia.com/] (Aiforia Technologies Oy, Helsinki, Finland) or the Wolution platform [https://console.wolution.ai/] (Wolution GmbH & Co. KG, Planegg, Germany). For both automated platforms, in case of flawed sections that could not be automatically analysed, the counts from the section before and after were averaged to prevent methodological bias (for

stereology, these sections were omitted, and this information was taken account for the calculation of the stereological estimates).

The Aiforia-platform was only used for analysis of Hematoxylin-counterstained DAB-stained brain sections. It uses a supervised training to establish a non-disclosed deep CNN algorithm that recognized TH-positive neurons, based on nuclear/cell morphology and TH signal. The Aiforia algorithm was trained with 4064 TH-positive neurons[54] and comprised two layers. The individual TH-positive cells were segmented in the first one and counted in the second one (see Penttinen, Parkkinen[74]). For generating the ground truth data used for algorithm training, one circle is placed on top of the nucleus within each TH-positive neuron (instead of marking the whole nucleus), similar as for our respective Wolution-based algorithms DLAP-3 and 4. Only TH-positive cells with a clearly visible/focused nucleus were considered for neuron counting, to avoid counting the same cell on more than one section. Processed images were uploaded on the Wolution platform. For counting of only DAB-stained TH-positive neurons (juvenile mice), DLAP-3 was used, for counting of hematoxylin counterstained TH-positive neurons (adult mice), DLAP-4 was utilised.

**Immunofluorescence, image acquisition, and analysis**. Immunofluorescence (IF) experimental procedures were performed essentially as previously described[54]. Mouse brains were perfused, stored and cut as already described for IHC. All washing and incubation steps were performed while shaking (300 rpm, microplate shaker, VWR).

For automated counting and relative signal intensity quantification analysis with the DLAP-5 algorithm, immunostaining was performed in five separate cohorts of mice, using sequential or simultaneous antibody incubation. 30 μm coronal midbrain free-floating sections were washed (three times in 1x PBS for 10 min). An additional antigen retrieval step was applied for D2- and Aldh1A1-antibodies (100 °C for 10 min in pH6 (D2) or 80 °C for 30 min in pH 9 (Aldh1A1). Sections were blocked for 2 h with 10% normal goat serum (NGS, Vector Laboratories) and 0.5% Triton X-100 in 1x PBS. After blocking, sections were incubated with rabbit or chicken anti-TH primary antibody (1:1000, Merck), and/or mouse anti-DAT (1:500, Thermo Fisher Scientific), chicken anti-calbindin d28K (1:1000, Novus bio), rabbit anti-DRD2 (1:200, Proteintech), rabbit anti-Aldh1a1 (1:200, Abcam) primary antibody in carrier solution (1% NGS, 0.5% Triton X-100 in 1x PBS) overnight at 4 °C. For subsequent primary antibody incubations, sections were washed (three times in 1x PBS for 10 min at RT) before the next overnight incubation at 4 °C. Sections were washed (three times in 1x PBS for 10 min at RT) and incubated with Alexa Fluor 488 goat anti-mouse secondary antibody (1:1000, Thermo Fisher Scientific), Alexa Fluor 647 goat anti-rabbit (1:500, Thermo Fisher Scientific), and Alexa Fluor 647 goat anti-chicken (1:1000, Thermo Fisher Scientific), in carrier solution for 3 h at RT in darkness. After washing (three times in 1x PBS for 10 min at RT), the sections were mounted on SuperFrost® Plus glass slides (VWR) with VectaMount Permanent Mounting Medium with DAPI (Vector Laboratories).

For relative quantification of fluorescence-signal intensities in cellular compartments (plasma-membrane, cytoplasm, nucleus) with DLAP-6, two similar Kv4.3 staining protocols were utilised, adapted from ref. [97], leading to very similar results. More precisely, one WT and one Kv4.3 KO brain (dataset 1) was processed, and images were acquired in the Goillard lab essentially as described[97]. The other brains (dataset 2) were perfused in the Goaillard lab, and further processed in the Liss lab, according to the following protocol. 50 μm free-floating

coronal midbrain sections were blocked for 1 h 30 min with 5% NGS and 0.3% Triton X-100 in 1x PBS, followed by incubation with a rabbit anti-Kv4.3 primary antibody (1:10000, Alomone Labs) together with chicken (G)/mouse (L) anti-TH primary antibody (1:1000, Abcam / Merck) in a carrier solution (1% NGS, 0.3% Triton X-100 in 1x PBS) overnight at 4 °C. Sections were then washed (three times in 0.3% Triton X-100 in 1x PBS for 15 min) and incubated with Alexa Fluor 488 goat anti-rabbit secondary antibody (1:1000, Thermo Fisher Scientific) and Alexa Fluor 546 goat anti-mouse secondary antibody (1:1000, Thermo Fisher Scientific) in carrier solution for 2 h at room temperature. After washing (three times in 0.3% Triton X-100 in 1x PBS for 15 min), sections were incubated with DAPI (1.5 μg/mL; Sigma-Aldrich) for 10 min, washed (two times in 1x PBS for 10 min), and mounted on glass slides using Vectashield mounting medium.

Fluorescent images for automated TH-IF positive neuron counts and relative signal intensity quantification (DAT-IF) were acquired with a Leica DM6 B epifluorescence microscope (Leica Microsystems) using a 63X oil objective. All image parameters were set using the LAS X software (Leica microsystems) to avoid saturated pixels, and identical acquisition settings were maintained (a prerequisite for relative signal intensity quantification). TH-IF was visualized at 546 nm (exposure = 200–500 ms, gain = 1–2), DAT-IF at 488 nm (exposure = 400–600 ms, gain = 1–2), CB-IF, D2-IF and Aldh1A1-IF at 660 nm (exposure = 300–500 ms, gain = 2) and DAPI at 405 nm (exposure = 60–300 ms, gain = 1). Fluorescence lamp illumination was kept at 30%. Images were acquired as tile scans from a single plane of focus. Merged images were exported as Leica Image File (LIF).

For relative quantification of fluorescence-signal intensities in cellular compartments, 63X confocal quality images were acquired using a Zeiss LSM780 with the settings defined in ref. [97] for dataset 1. In dataset 2, confocal images using 100X/1.4 NA oil objective were acquired with STEDYCON (Abberior Instruments) mounted on an Olympus BX53 upright microscope (Olympus LS), and image parameters were adjusted using the STEDYCON web-based user interface (Abberior Instruments). To avoid saturated pixels (a prerequisite for relative signal intensity quantification), photon counts from all channels were monitored in the photon count range indicator using the "fire blue" colour-map. This map indicates saturated image parts in blue over a yellow-red gradient for each individual channel. Fluorochromes were excited using an Argon laser, and the following wavelengths were used for excitation: TH-IF at 546 nm, Kv4.3-IF at 488 nm, and DAPI at 405 nm. The following acquisition parameters were adjusted at the STEDYCON user interface: Pixel size 78 nm; pinhole diameter 0.71 AU; one-directional line accumulation acquisition; laser power 10% for TH and Kv4.3, 12% for DAPI. Images were acquired from a single plane of focus and were exported as OsmAnd Binary Maps (OBF) files.

For automated counting of TH-IF positive neurons und and relative signal intensity quantification, images were converted to 8 bit RGB colour TIFF images. The Fiji software was used to label and cut out the sections of interest (~28 consecutive SN sections within the caudo-rostral axis, bilaterally), and the ROI was identified and strictly marked according to typical anatomical landmarks and the cartesian coordinates according to the mouse brain atlas for each section, similar as described for IHC/DAB-stained neuron counting[50,78]. Exemplary ROI marking is depicted in Fig. 7a and Fig. S5a. Processed images were uploaded on the Wolution platform and DLAP-5 (Wolution GmBH & Co. HG) was used for analysis. The algorithm automatically identified cell bodies and nucleus of both TH- (red channel) and DAPI-positive

(blue channel) cells within the ROI and directly quantified the mean IF signal intensity per area for TH, DAPI and the target genes (green channel, DAT; magenta channel, CB, D2, and Aldh1A1) in the cell body and nucleus. Background (BG) regions (for normalization of signal intensities) were taken from regions outside the ROI by application of individual IF intensity thresholds for each RGB channel (as described in post-training procedures). All identified SN cells were manually checked for the correct and clear identification of full cell bodies and nuclei. If cell bodies were not completely separated from each other, or if they were cropped, fragmented, very small, or did not show a full nucleus, these cells were excluded from further semi-quantitative analysis (~30% of identified TH-positive SN neurons, Fig. S5a, Table S12). This strategy was used, as our aim was analysis of TH and DAT co-expression and signal-quantification. For absolute quantification of all TH-positive (or DAT-positive) neurons, a count of neuronal nuclei is required (and possible), as described for DLAP-3/4.

Raw signal intensities of each cell were normalised to the determined BG signals from the respective section. The BG signals determined from each analysed section were then normalised to the mean BG value for each animal, and outliers were removed. The threshold for defining DAT-negative SN neurons was manually determined, and set as the mean normalised BG value for each mouse, plus 1.5 to 2.5 times SD of the DAT relative fluorescence (RF) signal (depending on the different signal intensities of the different mouse cohorts; Fig. S5c). Similarly, a threshold of mean normalised BG value for each mouse plus 1 time SD for D2, plus 4 times SD for CB, and plus 5 times SD for Aldh1A1 was used (Fig. S8, Table S15).

For the generation of anatomical 2D/3D maps, the R-computational environment[174] version 4.2.1 with the Tidyverse package[175], Plotly [https://plot.ly/] and RMarkdown[176] was used, and coordinates as well as DAT and TH signal intensities were systematically adjusted in order to plot the different cohorts and animals into one graph. 2D-Maps were generated by using and adjusting the x and y coordinates from each analysed TH-positive neuron, automatically provided by the DLAP-5 algorithm. The respective z-coordinates were derived according to the thickness of the individual consecutive back-to-back brain sections (30 μm, bregma: caudal to rostral; −3.9 to −2.7). Alternatively, as DLAP processes single images within a part of a z-stack of images and processes them slice by slice, the output-files of the algorithms are easily amended, so the user can reassemble the z-stack of segmentations afterwards, to analyse data in three-dimensions. Training a 3-dimensional CNN is also possible with the DLAP platform, but was not necessary here. For normalization of anatomical coordinates of individual animals, we applied animal and SN hemisphere dependent z-score normalization of x and y coordinates using the "scale ()" function in R. Thereby, the centre of each SN hemisphere (defined as the midpoint of the coordinate scale for all identified SN neurons on this hemisphere) was set as 0,0-value for both axes, a commonly used strategy[177]. We also utilised these adjusted x and y coordinates for defining the lateral SN region as starting at >1.5 scaled x-axis units lateral (i.e., >377.8 μm) from each SN hemisphere-centre (0,0; Fig. S6, Table S17). For generation of 2D and 3D gradient plots, we used the "scale ()" function to generate scaled DAT and TH RF values: similar as for the anatomical coordinates, the mean of the DAT and the TH RF intensity of all anaysed TH-positive neurons was set as 0-value and the deviation from the mean was plotted for each individual neuron, using a colour gradient. For better gradient-visualization, we excluded the highest values from the gradient-plots of (0.3% of all neurons for DAT, 0.005% for TH).

For relative quantification of fluorescence-signal intensities in cellular compartments with DLAP-6 algorithm, OBF images were converted to 8 bit RGB colour TIFF images using the Fiji software.

For manual analysis, images were further processed using Fiji. First, the "polygon selection" tool was used to manually delineate a continuous membrane ROI around the cell based on the Kv4.3 IF signal (green channel). The cytoplasm was marked according to the immunofluorescence signal for the cytoplasmic TH (red channel), and nucleus was marked according to the DAPI signal (blue channel). In the next step, the "measure" function (under the "analyse" toolbar) was used to calculate the mean RF signal intensity for each compartment, already normalised to the respective area. For automated analysis, the same high resolution RGB colour TIFF images were uploaded on the Wolution platform and analysed with DLAP-6. Each identified TH-positive neuron was segmented into "cytoplasm", "membrane" and "nucleus", based on the cytoplasmic TH and nuclear DAPI fluorescence signal (after training, the plasma-membrane was reliably identified without a third membrane-marker). BG signal intensities were determined from images acquired outside the region of analysis, using individual RGB channel thresholds, and were used to normalise signal intensities for both, manual and automated analyses. All images were manually checked for the correct identification of membrane, cytoplasm and nuclear compartments similar as performed for analysis with DLAP-5 and ~90% of cells were used for further quantitative analysis. In case of a non-continuous membrane compartment identification, the weighted mean (area) of RF signal intensity in all identified membrane ROIs from each cell was used for analysis.

**Statistics and reproducibility**. Data analysis and graphical illustrations were performed using GraphPad Prism 9 (GraphPad Software, Inc.), Adobe Illustrator CC2015.3 (Adobe Systems Software), Wolution (https://wolution.com/), and Fiji (https://imagej.net/Fiji) software. Statistical tests were performed with GraphPad Prism 9. Bayesian probabilistic model was conducted in RStan with the R package version 2.21.7[172]. Anatomical maps were plotted using R computational environment[174] version 4.2.1 with Tidyverse package[175], Plotly [https://plot.ly/], and RMarkdown[176].

Normal distribution was tested with D'Agostino-Pearson omnibus normality test. Correlations were performed using Pearson correlation tests, proportionality constants was calculated from the ratio between two directly corresponding values. In graphs, data are given as boxplots, showing median and whiskers representing 10th–90th percentile. For comparing one parameter only, non-parametric Mann-Whitney $U$ tests and Kruskal-Wallis tests, with post hoc Dunn's multiple comparison was used. For comparison between categorical %, Fisher's exact tests and Chi-squared tests were used, as indicated. Testings for two independent parameters were performed via two-way ANOVA with Tukey's multiple comparison tests. ROUT outlier test (Q = 1) was used to remove outliers. Statistical significances are indicated as ns > 0.05, $*p < 0.05$, $**p < 0.01$, $***p < 0.001$, and $****p < 0.0001$.

**Reporting summary**. Further information on research design is available in the Nature Portfolio Reporting Summary linked to this article.

## Data availability
Data used to generate figures in this manuscript, supporting analysis and RNAscope probe details can be found in the associated Supplementary Information file. The source data can be obtained from Supplementary Data 1. The link to our Deep Learning-based Analysis Platform is https://console.wolution.ai/. All original data and further information to interpret, verify and extend the research in this article are available from the corresponding authors on reasonable request.

## Code availability
Further information regarding the utilised algorithms and the individual codes are available upon request to the corresponding authors.

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

## Acknowledgements

We are particularly grateful to the brain donors and the German Brain Bank. We thank Matthias Bayerle and Dennis Kätzel for help with initial DAT-immunohistochemistry. This study was supported by the German DFG (LI-1745/1-3, GRK1789, SFB1506), the Austrian FWF (F44-12), the Alfried Krupp Foundation, the Boehringer Ingelheim Ulm University BioCenter (BIU), a Wellcome Trust Collaborative Award, and a Research Fellowship by the Hamburg Institute for Advanced Study (HIAS), all to B.L. NB and JB were supported by the International Graduate Schools of Molecular Medicine and of Aging (CEMMA) at Ulm University. MH was supported by an "Experimental Medicine" scholarship of Ulm University.

## Author contributions

N.B. carried out the RNAscope analysis. S.R. carried out the protein quantification in cellular compartments, and co-supervised M.H. and M.O. with neuron count and protein quantification in IF images. D.W. and M.M. implemented and customized all algorithms for automated analysis. S.M. performed IHC and IF staining, J.W. performed stereology and automated cell count analysis for juvenile mice, H.H. and J.B. for adult mice. J.R.F. and J.M.G. provided Kv4.3 KO mice, and high resolution Kv4.3 immunofluorescence images. M.F. implemented the mixed effects model. C.P. generated anatomical maps, J.D. and R.P. helped with supervision and analysis. B.L. designed and supervised the study and wrote the manuscript, together with N.B. and S.R. All authors revised the manuscript.

## Funding

## Competing interests

The authors declare no competing interests.

**Additional information**

