## [Peer Review File · Communications Biology]

Reviewers' comments:

Reviewer #1 (Remarks to the Author):

This paper presents the deep learning platform for quantifying neuronal images collected by different imaging systems. In this platform, six pre-trained deep networks can be directly used for mRNA, cellbody and nucleus segmentation. This platform also provides the solution to training these networks by using new datasets. The presented results suggest that the platform may be valuable for neuroscience research. In neuronal image analysis, numerous methods and networks have been published. However, it is difficult to use these methods in practice for the researchers who are unskilled in programming. The deep learning platform may alleviate this situation. In addition, building such a platform needs a lot of work. So, I recommend this work to be published. My main concerns are listed below.

1. From the networks descriptions and the presented results, it is difficult to judge that the platform can well separate the cell bodies that are distributed densely. In fact, accurate segmentation of dense cells is an important topic in biomedical image analysis. Cellpose provides a powerful tool for this purpose. Compared to Cellpose, the advantages of the platform should be discussed.
2. Large-volume imaging can provide accurate structural anatomical information and thus become more and more popular. The presented deep learning platform seems to only process two-dimensional images. This disadvantage constrains the practical application of the platform.
3. mRNA dots detection can be classified as small object detection. In general, accurate mRNA dots detection needs some special design in network or loss function. In this platform, how to design to achieve accurate detection.
4. The training set are not sufficient in which several thousands of cells are included at most. How to ensure the generalization performance of the pre-training networks.

Reviewer #2 (Remarks to the Author):

The authors present a technique for the automated analysis of mRNA expression in situ of prelabelled neurons. This technique is a perfect complement to current methods of single-cell mRNA expression analyses, which use dissociated neurons and are subject to variable neuronal survival during dissociation protocols and, most importantly, the information of anatomical localization of these cells is lost. As in similar techniques, the limitation of the technique presented here is the number of labels that can be used.

The authors focused on developing an automated approach of counting labelled neurons in fixed tissue slices, focusing on TH+ cells, and DAT (SLC6) expression. Combined with probe-derived fluorescence quantitation of Cav1.3 and Cav2.3 mRNA. During this process they identified TH-expressing cells, which did not express DAT.

Comments:

1. Abstract:

For clarity, I would recommend using "TH+" instead of "dopaminergic" on line 10 of abstract, as TH alone is not a proof of the cells being dopaminergic (Bjorklund & Dunnett 2007).

2. Are the DAT negative cells truly mDA neurons? Many other cell types express TH.

Since the region was marked manually, is there a possibility that this included a border region of anatomically different area? Can they be interneurons? (An., et al., 2021).

It would be useful to have at least one more marker and an image that shows the anatomical position of these cells, as well as its processes to show its overall shape, as only a cell body is visible.

3. Page 7, last paragraph, 2nd line: "in in"

4. The co-expression of TH and DAT was analyzed by Tiklova in her publication which looked at the expression of TH, DAT and other markers in Pitx-positive SN and VTA neurons. In that publication, the TH+/DAT-low neurons were identified as VTA Glutamatergic and VTA GABAergic neurons. The results of Tiklova illustrate the necessity of using more than two markers to identify the type of a neuron present.

5. The authors state that these cells cannot be part of the VTA due to their ventral-SN localization. It would be helpful if they supplied images to show the localization of these cells, as well as images that show the entire cell and its processes, so the neuroanatomy could be considered in their classification. Anderegg and Pulin (2015) showed that the VTA GABA neurons show a significant expression of CB, hence the authors may have identified a similar type of a neuron in the SN.

In summary, TH expression is not sufficient for the classification of a neuron as dopaminergic. Expression of TH is not even sufficient to prove that a neuron is catecholaminergic, let alone dopaminergic (Bjorklund & Dunnett 2007). The expression of TH does not mean that the neuron can produce and release dopamine. In order to classify a neuron as DA, one needs to consider a number of other markers, such as:

vesicular monoamine transporter SLC18a2 (VMAT-2)

DOPA decarboxylase (AADC/DDC)

Aldehyde Dehydrogenase 1 Family Member A1) ALDH1A1

Absence of Pax6 expression

Kv4.3 may be highly expressed in DA neurons, but it is not specific for DA neurons, as it is expressed by many different types of neurons. It is a useful marker to test the automated system, but it is not a good marker of DA neurons.

Page 16:

The authors state that several markers of mDA neurons are missing the the Th+DAT- cells, yet they still call these cells DA neurons. If all the major markers of an mDA neuron are missing, including Aldh1a1 and SOX6, perhaps their identity should not be assigned as mDA or dopaminergic at all. The expression of Vglut2 does not identify them as DA, as this gene is expressed in many types of neuronal cells. It is expressed at a particularly high level in VTA GABAergic neurons (Anderegg and Pulin, 2015).

References:

Björklund A, Dunnett SB. Dopamine neuron systems in the brain: an update. *Trends Neurosci.* 2007 May;30(5):194-202. doi: 10.1016/j.tins.2007.03.006. Epub 2007 Apr 3. PMID: 17408759.

Tiklová K, et al. Single-cell RNA sequencing reveals midbrain dopamine neuron diversity emerging during mouse brain development. *Nat Commun.* 2019 Feb 4;10(1):581

Anderegg A, Poulin J-F, and Awatramani R. Molecular Heterogeneity of Midbrain Dopaminergic Neurons - Moving Toward Single Cell Resolution. *FEBS Lett.* 2015 Dec 21;589(24 Pt A):3714-26.

An S, Li X, Deng L, Zhao P, Ding Z, Han Y, Luo Y, Liu X, Li A, Luo Q, Feng Z, Gong H. A Whole-Brain Connectivity Map of VTA and SNc Glutamatergic and GABAergic Neurons in Mice. *Front Neuroanat.* 2021 Dec 23;15:818242.

Reviewer 1:

This paper presents the deep learning platform for quantifying neuronal images collected by different imaging systems. In this platform, six pre-trained deep networks can be directly used for mRNA, cellbody and nucleus segmentation. This platform also provides the solution to training these networks by using new datasets. The presented results suggest that the platform may be valuable for neuroscience research. In neuronal image analysis, numerous methods and networks have been published. However, it is difficult to use these methods in practice for the researchers who are unskilled in programming. The deep learning platform may alleviate this situation. In addition, building such a platform needs a lot of work. So, I recommend this work to be published. Main concerns are listed below.

We are very pleased that reviewer 1 recommends our work to be published, and we are grateful for her/his considerate comments that helped us to further improve our manuscript as specified below.

1. From the networks descriptions and the presented results, it is difficult to judge that the platform can well separate the cell bodies that are distributed densely. In fact, accurate segmentation of dense cells is an important topic in biomedical image analysis. Cellpose provides a powerful tool for this purpose. Compared to Cellpose, the advantages of the platform should be discussed.

We fully agree with the reviewer that proper segmentation of the cell bodies is crucial, especially for dense cells. Thus, we have empirically ensured that our neural network generates a high quality segmentation, which acts as a well-suited basis to separate cells by application of conventional post-processing methods (such as the watershed algorithm) as specified in Table 2 and methods. We found that the combination of deep learning and conventional methods leads to very good separation results, as we now discuss in more detail in the revised manuscript (page 14).

We had not systematically compared our approach with “Cellpose 2.0” (<https://www.cellpose.org/>), a novel anatomical segmentation algorithm by Stringer and Pachitariu (Pachitariu & Stringer, 2022), but we have tested this approach now: Cellpose2.0 did enable equally proper segmentation of TH-positive neurons from our IF-images, as our DLAP algorithms. However, in contrast to DLAP, it does not enable any automated post-processing or post-analysis of segmented ROIs for further downstream analysis (like watershed algorithm, threshold based background exclusion, or quantification of fluorescent dot numbers or intensities) in one single package, but the user would require to write additional scripts for such downstream-applications. Hence, as Cellpose2.0 relies on Python packages, despite its Graphical User Interface, basic programming knowledge is required, e.g. for troubleshooting package related issues or run-time errors. Our DLAP approach directly combines deep learning based segmentation with more conventional post-processing methods (such as the watershed algorithm etc., see Table 2), as we found that the combination of deep learning and conventional methods lead to high quality results. One particular additional advantage of our DLAP-6 is that it automatically segments the membrane compartment in the absence of a membrane marker. We now discuss these advantages of DLAP in more detail in the revised manuscript (page 14-15).

2. Large-volume imaging can provide the accurate structural anatomical information and thus become more and more popular. The presented deep learning platform seems to only process two-dimensional images. This disadvantage constrains the practical application of the platform.

We fully agree with the reviewer. Our platform currently allows analyses of single images within a part of a z-stack of images and processes them slice by slice. As the output-files of the algorithms are easily amended, the user can reassemble the z-stack of segmentations afterwards, to analyze data in three dimensions. For our extended DAT co-expression analysis (we doubled the number of mice to N=14), we now did add anatomical 3D-maps, in addition to the 2D-maps, for better visualization of the DAT-negative SN neuron location, and of the DAT- and TH-gradients, by deriving the z-coordinates from the thickness of the individual consecutive brain sections (new online Figure S7, interactive HTML-file). An alternative approach that is feasible with our DLAP platform is to train a 3-dimensional CNN. We describe these approaches now in the revised manuscript (pages 31-32).

3. mRNA dots detection can be classified as small object detection. In general, accurate mRNA dots detection needs some special design in network or loss function. In this platform, how to design to achieve accurate detection.

Again, we fully agree with the reviewer, and indeed we found that for small objects (i.e. mRNA-

derived fluorescence dots), the pre-trained Deeplab3 algorithm delivered rather poor results. This is why we developed the much simpler FCN algorithm for RNAScope dot detection (DLAP-1 & -2), which has a smaller receptive field and conserves image resolution at each convolutional layer (Long et al, 2015). FCN algorithms perform better for small object detection, and consequently, they detected the RNAScope derived fluorescence dots much better than Deeplab3 (see Tables 1, S1, and page 18, 19 in the revised manuscript). For the loss function (Rajaraman et al, 2021) we briefly experimented with training the FCN with the dice loss instead of the common pixel-wise cross-entropy (Yeung et al, 2022), but did not find a significant improvement. We discuss this now in the revised manuscript (page 14).

4. The training set are not sufficient in which several thousands of cells are included at most. How to ensure the generalization performance of the pre-training networks.

We ensured performance of the pre-trained networks in the following way: We used 80% of manually labelled images for the training data set, and the remaining 20% of the images for the “test data” set. Train-test-split on the level of images was chosen to avoid data leakage. We trained each algorithm with the training data set until the training error/loss function converged to a minimum. We used “Early Stopping” in the training process to avoid overtraining. We verified that we did not overtrain, by tracking the pixel error in the “test data” (training was stopped as soon as the pixel error was getting worse not better). Performances of all algorithms were evaluated on the respective “test data” (compare Table 2).

If the concern of the reviewer is generalization to other images (for example using a different imaging system), the platform allows easy retraining of the algorithms on such data, as training and image-analyses are carried out via an easy-to-use web-based interface. The FCN algorithms are not pre-trained, but can be easily trained from scratch, as less training data is required, because of its smaller size compared to the Deeplab neural network.

We explain this now in more detail in the revised manuscript (pages 14, 19-21).

Reviewer 2:

The authors present a technique for the automated analysis of mRNA expression in situ of prelabelled neurons. This technique is a perfect complement to current methods of single-cell mRNA expression analyses, which use dissociated neurons and are subject to variable neuronal survival during dissociation protocols and, most importantly, the information of anatomical localization of these cells is lost.

We are very grateful that reviewer 2 evaluates our approach as “a perfect complement to current methods”, and for his/her very helpful comments and advice to further improve our manuscript.

As in similar techniques, the limitation of the technique presented here is the number of labels that can be used. The authors focused on developing an automated approach of counting labelled neurons in fixed tissue slices, focusing on TH+ cells, and DAT (SLC6) expression. Combined with probe-derived fluorescence quantitation of Cav1.3 and Cav2.3 mRNA. During this process they identified TH-expressing cells, which did not express DAT.

The reviewer is correct. However, we would like to note that DLAP approaches 5 & 6 are optimized for immunofluorescence-derived signals and protein-detection, not mRNA (compare Table 2).

1. Abstract: For clarity, I would recommend using “TH+” instead of “dopaminergic” on line 10 of abstract, as TH alone is not a proof of the cells being dopaminergic (Björklund & Dunnett, 2007). 2. Are the DAT negative cells truly mDA neurons? Many other cell types express TH.

We fully agree with the reviewer, we changed the abstract accordingly, and we added a respective sentence to the discussion (page 15: ...*TH-expression alone is not a proof of neurons being dopaminergic (Björklund & Dunnett, 2007)*). In addition, we write now TH-positive (DAT-negative) SN neurons, instead of SN DA neurons throughout the manuscript.

Since the region was marked manually, is there a possibility that this included a border region of anatomically different

area? Can they be interneurons? (An et al, 2021)

We strictly followed our well-established anatomical landmarks, and cartesian coordinates according to the mouse brain atlas for marking the SN area (Paxinos & Keith B. J. Franklin, 2007), and we have now double-checked all sections analyzed, to ensure that we only marked the SN, accordingly. We have clarified this in the revised manuscript (page 31-32 and Figures 7a, S5a and S6).

TH-positive but DAT negative striatal interneurons have been described (Xenias et al, 2015). However, interneurons in the SN are rare (Brown et al, 2014; Smith & Masilamoni, 2010) and rather small (~ three times smaller than classical SN DA neurons, Liss et al, 1999). Given the still relatively large cell-body size of the DAT-negative lateral SN neurons (~120 μm^2 compared to ~195 μm^2), here we would not unequivocally conclude that these are interneurons, while we do not rule out this possibility. We now discuss this point in the revised manuscript (page 15).

It would be useful to have at least one more marker and an image that shows the anatomical position of these cells, as well as its processes to show its overall shape, as only a cell body is visible.

4. The co-expression of TH and DAT was analyzed by Tiklova in her publication which looked at the expression of TH, DAT and other markers in Pitx-positive SN and VTA neurons. In that publication, the TH+/DAT-low neurons were identified as VTA Glutamatergic and VTA GABAergic neurons. The results of Tiklova illustrate the necessity of using more than two markers to identify the type of a neuron present.

5. The authors state that these cells cannot be part of the VTA due to their ventral-SN localization. It would be helpful if they supplied images to show the localization of these cells, as well as images that show the entire cell and its processes, so the neuroanatomy could be considered in their classification. Anderegg et al (2015) showed that the VTA GABA neurons show a significant expression of CB, hence the authors may have identified a similar type of a neuron in the SN.

We fully agree with the reviewer that analysis of additional DA neuron markers, of the neuronal morphology, and better visualization of their anatomical localization is very helpful to further characterize the DAT-negative SN neuron population. We have now substantially extended our analysis of DAT co-expression in TH-positive SN neurons by adding the data for seven additional mice, for three additional markers (D2, CB, Aldh1A1), and by additional analyses, like providing cell-body sizes and anatomical 3D-maps (new Figures 7-9, S6, and Tables S12-S17):

For better visualization and evaluation of the DAT-negative SN neuron location, and of the DAT- and TH-gradients, over the whole caudo-rostral axis of the SN, we added new anatomical 3D-maps (new online Figure S7 - interactive HTML-file). We also have now added further high-resolution original pictures (Figures 7a, 10, and S5a). In addition, we have added (to Figures 7-9, 11, S5-S7) schematic exemplary sagittal and coronal mouse brain sections (modified from Paxinos et al, 2007), illustrating the analyzed caudo-rostral extend of the SN (bregma: -3.9 to -2.7), and the definition and location of its lateral parts in coronal sections.

To address morphology, we now also systematically compare cell-body sizes, revealing that the DAT-negative neurons in the lateral SN are ~40% smaller compared to classic SN DA neurons (new Figure 7e, new Table S17). After having analyzed the morphology of DAT-negative and DAT-positive SN neurons by hand (10-15 neurons/animal), we gained evidence that the dendritic organization and orientation might also differ between them (as illustrated in Figure 10). To quantify automatically dendritic numbers and orientations, we have now extended our algorithm, and we are currently training and further optimizing it. As this turned out to be a more complex issue, we hope the reviewer agrees that this will be presented in a follow-up study.

Most importantly, we added two completely new, full data-sets, detailing the co-expression analysis of two additional markers for dopaminergic subpopulations: the dopamine D2-autoreceptor and the aldehyde dehydrogenase Aldh1A1 (new Figures 10, 11, S8, and Tables S14-S17). These novel data include co-expression maps for TH, DAT, and for D2-AR (a marker for classical SN DA neurons), for CB (a marker for less vulnerable SN DA neurons), and for Aldh1A1 (a marker for highly vulnerable SN DA neurons), respectively. In a nutshell: compared to DAT-positive SN neurons, the DAT-negative neurons in the lateral SN are immune-negative for Aldh1A1 (~3% vs. 62%) co-express CB to much higher degree (~67% vs 8%), and co-express D2-AR to a lower degree (~70% vs 93%). Our single nuclei RNA sequencing data (also part of a follow-up study) confirmed our findings for DAT, Aldh1A1, D2, and CB at the mRNA-level, and they indicate that the DAT-negative SN neurons are Vglut2- and VMAT2-positive (with lower VMAT2-levels), and Gad1/2- and Sox6-negative (SOX6 is at least significantly reduced). Due to the localization of the DAT-negative neurons predominantly

in the very lateral parts of the SN, we can exclude that they are VTA neurons (see above). We discuss this now on page 15/16. We would also rather propose that they are not similar to the VTA GABAergic neurons described in Anderegg 2015, as those neurons were DAT-positive and our DAT-negative neurons are presumably non-GABAergic (GAD1/2-negative).

In summary, TH expression is not sufficient for the classification of a neuron as dopaminergic. Expression of TH is not even sufficient to prove that a neuron is catecholaminergic, let alone dopaminergic (Bjorklund & Dunnett 2007). The expression of TH does not mean that the neuron can produce and release dopamine. In order to classify a neuron as DA, one needs to consider a number of other markers, such as: vesicular monoamine transporter SLC18a2 (VMAT-2), DOPA decarboxylase (AADC/DDC), Aldehyde Dehydrogenase 1 Family Member A1) ALDH1A1, Absence of Pax6 expression. Kv4.3 may be highly expressed in DA neurons, but it is not specific for DA neurons, as it is expressed by many different types of neurons. It is a useful marker to test the automated system, but it is not a good marker of DA neurons. Page 16: The authors state that several markers of mDA neurons are missing the the Th+DAT-cells, yet they still call these cells DA neurons. If all the major markers of an mDA neuron are missing, including Aldh1a1 and SOX6, perhaps their identity should not be assigned as mDA or dopaminergic at all. The expression of Vglut2 does not identify them as DA, as this gene is expressed in many types of neuronal cells. It is expressed at a particularly high level in VTA GABAergic neurons (Anderegg and Pulin, 2015).

As detailed above, we agree with the reviewer, and we have substantially extended our data-sets, now provide results for additional markers, and we have revised the manuscript accordingly (highlighted in red), as specified above.

We also fully agree that the ion channel Kv4.3 is not a marker for DA neurons, but it is highly expressed in the plasma-membranes of SN DA neurons (Haddjeri-Hopkins et al, 2021; Liss et al, 2001), it is involved in defining their vulnerability (Dragicevic et al, 2015; Subramaniam et al, 2014), and a very good antibody is available for this protein. Hence, we used Kv4.3 as marker for the plasma-membranes of SN DA neurons, in order to train the algorithm to detect and separately analyze signals in plasma-membranes of DA neurons. We have clarified this in the revised manuscript (page 12).

3. Page 7, last paragraph, 2nd line: "in in": We corrected this mistake.

References:

- An S, Li X, Deng L, Zhao P, Ding Z, Han Y, Luo Y, Liu X, Li A, Luo Q, Feng Z, Gong H (2021) A Whole-Brain Connectivity Map of VTA and SNc Glutamatergic and GABAergic Neurons in Mice. *Frontiers in neuroanatomy* **15**: 818242
- Anderegg A, Poulin J-F, Awatramani R (2015) Molecular heterogeneity of midbrain dopaminergic neurons – Moving toward single cell resolution. *FEBS Letters* **589**: 3714-3726
- Björklund A, Dunnett SB (2007) Dopamine neuron systems in the brain: an update. *Trends in neurosciences* **30**: 194-202
- Brown J, Pan W-X, Dudman JT (2014) The inhibitory microcircuit of the substantia nigra provides feedback gain control of the basal ganglia output. *eLife* **3**: e02397
- Dragicevic E, Schiemann J, Liss B (2015) Dopamine midbrain neurons in health and Parkinson's disease: Emerging roles of voltage-gated calcium channels and ATP-sensitive potassium channels. *Neuroscience* **284**: 798-814
- Haddjeri-Hopkins A, Tapia M, Ramirez-Franco J, Tell F, Marqueze-Pouey B, Amalric M, Goillard JM (2021) Refining the Identity and Role of Kv4 Channels in Mouse Substantia Nigra Dopaminergic Neurons. *eNeuro* **8**
- Liss B, Franz O, Sewing S, Bruns R, Neuhoff H, Roeper J (2001) Tuning pacemaker frequency of individual dopaminergic neurons by Kv4.3L and KChip3.1 transcription. *The EMBO journal* **20**: 5715-5724
- Long J, Shelhamer E, Darrell T (2015) Fully Convolutional Networks for Semantic Segmentation. *Proc Cvpr Ieee*: 3431-3440
- Pachitariu M, Stringer C (2022) Cellpose 2.0: how to train your own model. *Nat Methods* **19**: 1634-1641
- Paxinos G, Keith B. J. Franklin M (2007) *The Mouse Brain in Stereotaxic Coordinates*: Elsevier Science.
- Smith Y, Masilamoni JG (2010) Substantia Nigra. In *Encyclopedia of Movement Disorders*, Kompoliti K, Metman LV (eds), pp 189-192. Oxford: Academic Press
- Subramaniam M, Althof D, Gispert S, Schwenk J, Auburger G, Kulik A, Fakler B, Roeper J (2014) Mutant α -synuclein enhances firing frequencies in dopamine substantia nigra neurons by oxidative impairment of A-type potassium channels. *The Journal of neuroscience : the official journal of the Society for Neuroscience* **34**: 13586-13599
- Xenias HS, Ibáñez-Sandoval O, Koós T, Tepper JM (2015) Are Striatal Tyrosine Hydroxylase Interneurons Dopaminergic? *The Journal of Neuroscience* **35**: 6584-6599

REVIEWERS' COMMENTS:

Reviewer #1 (Remarks to the Author):

The authors have answered my questions properly. I agree its publication in its current form.

Reviewer #2 (Remarks to the Author):

I examined the authors' responses and I am satisfied that the authors have addressed these fully and thoroughly. I have no further comments but to congratulate the authors on their contribution to current advances in research techniques.